# Deciphering a critical role of uterine epithelial SHP2 in parturition initiation at single cell resolution

Meng Liu[1,2,3,6], Mengjun Ji[1,6], Jianghong Cheng[1], Yingzhe Li[1], Yingpu Tian[1], Hui Zhao[2,3], Yang Wang[2,3], Sijing Zhu[2,3], Leilei Zhang[2,3], Xinmei Xu[2,3], Gen-Sheng Feng[4], Xiaohuan Liang[5], Haili Bao[2,3], Yedong Tang[2,3], Shuangbo Kong ![ORCID][2,3], Jinhua Lu ![ORCID][2,3], Haibin Wang ![ORCID][2,3] ✉, Zhongxian Lu ![ORCID][1] ✉ & Wenbo Deng ![ORCID][2,3] ✉

The timely onset of female parturition is a critical determinant for pregnancy success. The highly heterogenous maternal decidua has been increasingly recognized as a vital factor in setting the timing of labor. Despite the cell type specific roles in parturition, the role of the uterine epithelium in the decidua remains poorly understood. This study uncovers the critical role of epithelial SHP2 in parturition initiation via COX1 and COX2 derived PGF2α leveraging epithelial specific *Shp2* knockout mice, whose disruption contributes to delayed parturition initiation, dystocia and fetal deaths. Additionally, we also show that there are distinct types of epithelium in the decidua approaching parturition at single cell resolution accompanied with profound epithelium reformation via proliferation. Meanwhile, the epithelium maintains the microenvironment by communicating with stromal cells and macrophages. The epithelial microenvironment is maintained by a close interaction among epithelial, stromal and macrophage cells of uterine stromal cells. In brief, this study provides a previously unappreciated role of the epithelium in parturition preparation and sheds lights on the prevention of preterm birth.

For the initiation of parturition to occur, appropriate maternal decidua maturation, uterine contractility, cervical dilatation, and rupture of the chorioamniotic membranes are required[1]. Disruption of these interactions can lead to several diseases, such as preterm birth (PTB), delayed parturition, and dystocia. PTB, defined as birth earlier than 37 weeks of gestation in humans, is the leading cause of neonatal deaths, affecting about 10% of pregnancies with an estimated 1 million babies born premature every year worldwide[2]. In contrast, delayed parturition or dystocia, missing the normal parturition window ascribed to inadequate uterine contractions, failed parturition initiation, fetal malposition, or cephalopelvic disproportion, is also detrimental for the fetus and mother[3]. However, the strategies to diagnose or prevent the occurrence of these diseases are very limited. Therefore, it is urgently necessary to illustrate the underlying mechanisms of parturition.

There is increasing evidence suggesting the maternal decidua determines the timing of labor through a "decidual clock" by serving as a "signal hub" to coordinate the communication between mother and

[1]State Key Laboratory of Cellular Stress Biology, School of Pharmaceutical Sciences, Faculty of Medicine and Life Sciences, Xiamen University, Xiamen, Fujian, China. [2]Fujian Provincial Key Laboratory of Reproductive Health Research, Department of Obstetrics and Gynecology, The First Affiliated Hospital of Xiamen University, School of Medicine, Xiamen University, Xiamen, Fujian 361102, China. [3]State Key Laboratory of Vaccines for Infectious Diseases, Xiang An Biomedicine Laboratory, School of Medicine, Xiamen University, Xiamen, Fujian, China. [4]Department of Pathology, Division of Biological Sciences, University of California San Diego, La Jolla, CA, USA. [5]College of Veterinary Medicine, South China Agricultural University, Guangzhou 510642, China. [6]These authors contributed equally: Meng Liu, Mengjun Ji. ✉e-mail: haibin.wang@vip.163.com; zhongxian@xmu.edu.cn; wbdeng@xmu.edu.cn

fetus[4–7]. The decidua is a highly heterogeneous tissue, consisting of stromal cells, epithelial cells, endothelial cells, and immune cells[8]. Among them, *p53*, transformation-related protein 53 (*Trp53*), plays a critical role in parturition through the COX2/PGF2α pathway in decidua[9]. Further studies have unveiled its role in uterine decidual senescence by calibrating the balance between mTORC1 and AMPK signaling pathway[10,11]. Endothelial TLR4 (Toll-like receptor 4) in decidual bed senses inflammation signals by modulating anti-inflammatory pathway to overcome inflammation-induced PTB[7]. In addition, genetic evidence also unveils that macrophages, neutrophils or dendritic cells are critical in combating inflammatory induced PTB via TLR4[12]. Apart from the essential role of these aforementioned cell types in parturition, epithelial cells, which is vital nutrition source in both human and mouse, undergo dynamic reorganization at later stage of pregnancy[13–15], while its role in parturition initiation remain scantly appreciated.

SHP2, encoded by the *Ptpn11* gene, is an ubiquitously expressed nonreceptor protein tyrosine phosphatase that participates in cell migration, growth, death and signal transduction, including JAK-STAT3, PI3K-AKT, NF-κB and RAS-RAF-ERK signaling pathways[16]. The null mutation of *Shp2* causes blastocyst lethality with inner cell mass death, decreased trophoblast giant cells, and failure to yield trophoblast stem cell lines due to defective activation of SRC/RAS/ERK pathway[17]. Uterine-specific deletion of *Shp2* by PR-Cre or AMHR2-Cre shows compromised embryo implantation or decidualization, respectively[18,19]. However, the role of epithelial *Shp2* in parturition initiation remains elusive.

In the current study, we unravel the previously unappreciated role of epithelium in parturition initiation leveraging genetic, molecular and pharmacological approaches. Epithelial specific deletion of *Shp2* contributes to failed parturition initiation, leading to dystocia and augmented fetal death due to reduction of epithelial COX1 (Cyclooxygenase 1) and COX2 (Cyclooxygenase 2) expression and ultimately resulting in a decreased level of PGF2α. In addition, our results herald the important role of epithelial *Shp2* in epithelium growth and uncover the extensive communications between epithelium and stroma. Collectively, our studies elucidate the role of epithelial *Shp2* in regulating PGF2α level and decidua at term to determine the onset of parturition.

## Results

### Uterine epithelium *Shp2* deficiency contributes to delayed parturition

To dissect out the physiological role of epithelial SHP2 in parturition, we generated a genetic mouse with uterine epithelium specific deletion of *Shp2* by crossing *Shp2*-loxp mice (*Shp2*^f/f^) with *Ltf*-iCre mice (*Ltf*^tCre/iCre^). Double immunofluorescence of CK8 and SHP2 showed that SHP2 was efficiently deleted in *Ltf*^tCre/iCre^ *Shp2*^f/f^ (*Shp2*^d/d^ hereafter) mice at day 19 (Fig. 1a). It was interesting that *Shp2*^d/d^ female mice experienced significantly delayed onset of labor and dystocia with lower maternal survivor rates (61%) (Fig. 1b, c). The comparable number of implantation sites and similar localization and expression of COX2, an early marker for embryo implantation, in both genotypes indicated limited influence of epithelial SHP2 in embryo implantation (Supplementary Fig. 1a–c). This evidence indicated the critical role of SHP2 in parturition initiation.

### Epithelial *Shp2* directs epithelial COX1/COX2 expression and PGF2α production

The regulation of parturition involves ovarian steroid hormones, progesterone (P4) and estrogen (E2)[20]. We measured the levels of P4 and E2 in day 19 serum and found higher P4 and lower E2 levels in *Shp2*^d/d^ mice (Fig. 1d, e). To further investigate the underlying mechanism of these irregular hormone levels, we detected the expression of enzymes for hormone metabolism in the ovary. Our results showed that 20α-HSD (encoded by *Akr1c18*), the enzyme for P4

degradation, was significantly lower in corpus luteum of the ovary in *Shp2*^d/d^ mice (Fig. 1f). Meanwhile, STAR (Steroidogenic acute regulatory protein), which transports cholesterol from the outer mitochondrial membrane to the inner mitochondrial membrane, was higher in corpus luteum of *Shp2*^d/d^ mice (Fig. 1g). These observations strongly suggested that ablation of epithelial *Shp2* disturbed the onset of parturition due to abnormal hormone production in the ovary.

Since luteolysis is under the regulation of decidual PGF2α[21], we investigated the expression of COX2, a rate-limiting enzyme responds for PG production by catalyzing prostaglandins produced from arachidonic acid, in day 19 uteri. COX2 was downregulated before labor in *Shp2*^d/d^ mice (Fig. 1h), which led us to investigate local prostaglandin levels. There was significantly lower PGF2α level on day 19 in *Shp2*^d/d^ mice (Fig. 1i). Considering *Cox1*^-/-^ mice had delayed parturition with increased neonatal death and impaired luteolysis[22,23], which recapitulate similar reproductive phenotypes observed in *Shp2*^d/d^ mice, we also detected the expression and localization of *Ptgs1* in uteri of both genotypes. It was interesting to notice that *Ptgs1* was specifically localized in epithelium and was overtly decreased in *Shp2*^d/d^ epithelium (Fig. 1j, k and Supplementary Fig. 2a). *Akr1b3* (Aldo-keto reductase 1b3), one of the enzymes responsible for PGF2α synthesis in conjunction with with COX1, was also downregulated in *Shp2*^d/d^ epithelium(Fig. 1l and Supplementary Fig. 2b–c). These results indicated that insufficient local PGF2α production contributes to parturition delay in the absence of epithelial SHP2. To interrogate this hypothesis, PGF2α (50 μg/mice) was injected on day 19 in *Shp2*^d/d^ mice. Compared with *Shp2*^d/d^ mice injected with PBS, more mice delivered earlier in the PGF2α injection group with significantly increased survival rates of the mother (83% vs 40%) (Fig. 1m, n). This decreased PGF2α production was also reflected in the blunt expression of *Oxtr* (Oxytocin receptor) in the muscle, downstream of PGF2α, in SHP2 epithelial ablated uterus (Supplementary Fig. 3). These results indicated that epithelial *Shp2* directed parturition by regulating local PGF2α production via both COX1 and COX2, which ultimately initiated parturition.

### Epithelium heterogeneity approach parturition

Above results indicate that epithelium is critical for parturition initiation. Previously studies reveal that epithelium undergoes dynamic change from day 1 to 8[24], while the role and characterization of epithelium at peri-parturition stage remains largely elusive. To depict the cell heterogeneity and gene expression profiling of epithelium approach parturition, mouse uteri on day 16 (non-labor stage) and day 19 (peri-labor stage) were collected. After the removal of decidua, placenta and fetus, the remaining uterine tissues were subjected to single cell RNA sequencing (scRNA-seq) (Fig. 2a). There were a total of 23 distinct clusters, including four different epithelial cells (Epi), three stromal cells (Str), endothelial cells (Endo), two muscle cells (Mus), mesothelial (Meso), pericytes (Peri), and immune cells (Macrophages, NK cells, T cells, Monocytes, Mast cells) as annotated based on their respective maker genes (Fig. 2b–d).

There were four different epithelial cells in our scRNA-Seq, inclduing two luminal epithelial cells (LE_0 and LE_5), gland (Gland_10) and proliferative epithelial cell (prolifEpi_20). The immunostaining of E-CAD (E-Cadherin) and CK8 (Cytokeratin 8) showed that there were two different luminal epithelial cells at implantation sites of later stage: columnar luminal epithelium lining the stroma close to muscle layer and squamous luminal epithelium close to the placenta and fetus as reported before (Fig. 2e and Supplementary Fig. 4)[15]. The staining of ERα (Estrogen receptor alpha) corroborated that only columnar luminal epithelial cells were ERα positive (Supplementary Fig. 5a–b). In addition, our result found that *Ezr* (Ezrin) was another specific marker to distinguish these two different epithelial cells with higher expression in columnar luminal epithelial cells (Fig. 2g and Supplementary Fig. 5b). We also identified that *Napsa* (Napsin A aspartic peptidase), a member of the peptidase A1 family of aspartic proteases, and *Csf1*

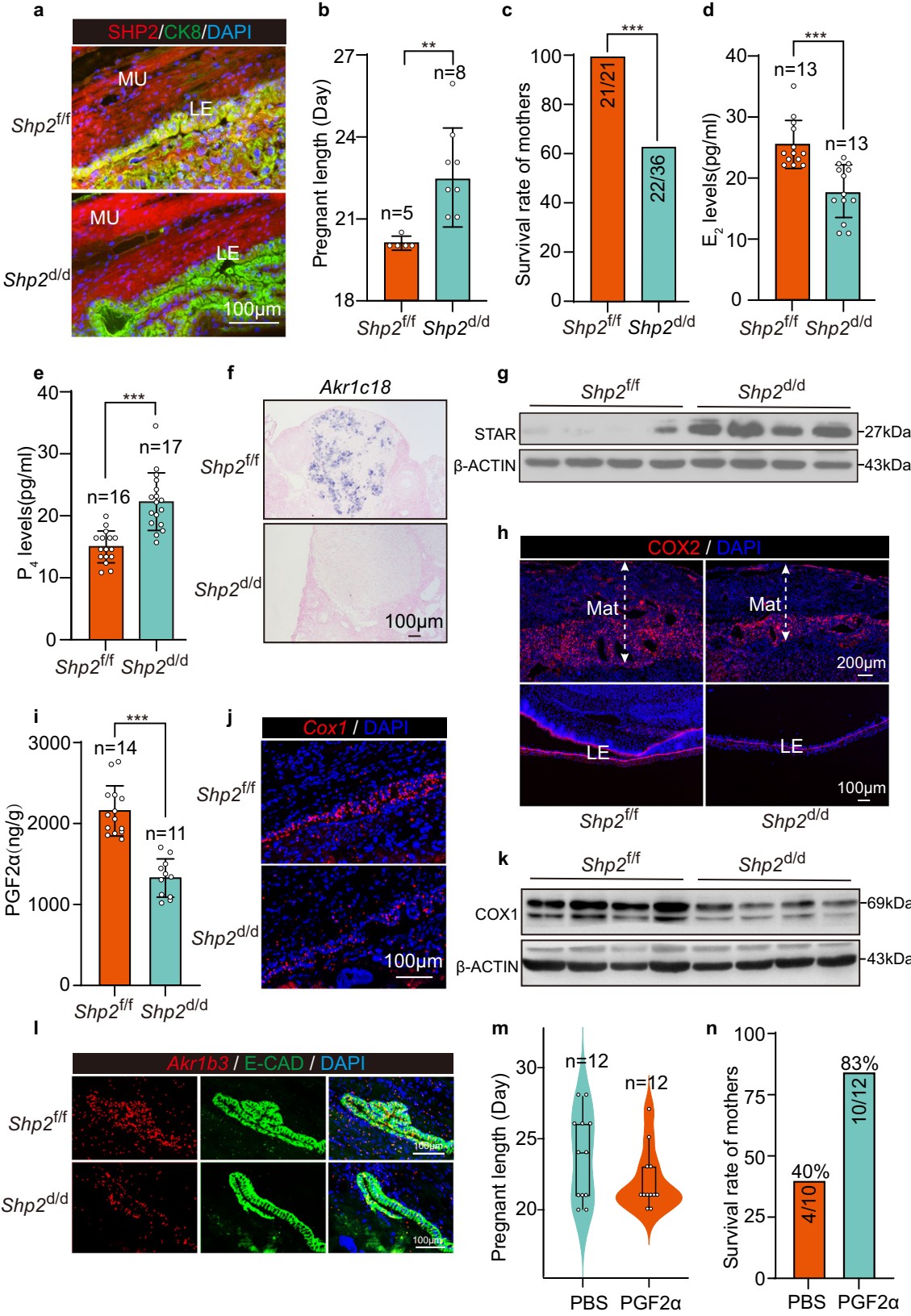

(Colony stimulating factor 1) specificall expressed in luminal epithelial cells in both genotypes (Fig. 2f and Supplementary Fig. 6a–b). Our previous work utilizing tissue clearing and 3D imaging showed that as the rapidly growing of embryo, glands become densely enriched at inter-implantion site on day 8 of pregnancy[24].Our current study also revealed that there were numerous glandular epithelial cells located in inter-implantation sites toward the anti-mesometrial side of the uteri (Supplementary Fig. 7).

Luminal and glandular epithelium possess distinct role in uteri. In order to globally portray the signature of luminal and glandular epithelial cells before labor, we investigated the biological function of each cell cluster by pathway enrichment analysis (Fig. 2g, h). The

**Fig. 1 | Uterine epithelial-specific deletion of *Shp2* leads to female dystocia with abnormal hormone levels and PGF2α synthesis. a** Coimmunostaining of SHP2 and CK8 in *Shp2*^f/f and *Shp2*^d/d mouse uteri on day 19. LE luminal epithelium, MU muscle. **b** Pregnant length in *Shp2*^f/f (*n* = 5) and *Shp2*^d/d (*n* = 8) female mice. Data represent as mean ± SEM. Two-tailed unpaired student's *t*-test, **\*\**p* = 0.0014. Each plot represents an individual sample. **c** Survival rates of pregnant *Shp2*^f/f and *Shp2*^d/d mother. The numbers in columns indicate survival pregnant females over total number of pregnant females. **\*\*\**p* = 8.46e−04, Fisher's exact test. **d** Serum estradiol-17β (E2) level in *Shp2*^f/f (*n* = 13) and *Shp2*^d/d (*n* = 13) mice on day 19. Data represent as mean ± SEM. Two-tailed unpaired student's *t*-test, **\*\*\**p* = 5.41e−05. **e** Serum progesterone (P4) level in *Shp2*^f/f (*n* = 16) and *Shp2*^d/d (*n* = 17) mice on day 19. Data represent as mean ± SEM. Two-tailed unpaired student's *t*-test, **\*\*\**p* = 5.15e−06. **f** In situ hybridization of *Akr1c18* in *Shp2*^f/f and *Shp2*^d/d ovaries on day 19. This result was repeated independently in three individual mice with similar results. **g** Western blot of STAR levels in *Shp2*^f/f and *Shp2*^d/d ovaries on day 19. β-ACTIN was used as a loading control, *n* = 4. **h** Immunofluorescence of COX2 in decidua and epithelium of *Shp2*^f/f and *Shp2*^d/d mice on day 19. Mat maternal tissue, LE luminal epithelum. **i** PGF2α level in *Shp2*^f/f (*n* = 14) and *Shp2*^d/d (*n* = 11) mouse uteri on day 19. Data represent the mean ± SEM. Two-tailed unpaired student's *t*-test, **\*\*\**p* = 1.89e-07. Each plot represents an individual sample. **j** Sm-FISH of *Cox1* in *Shp2*^f/f and *Shp2*^d/d mice on day 19. This result was repeated independently in three individual mice with similar results. **k** Western blot of COX1 levels in *Shp2*^f/f and *Shp2*^d/d ovaries on day 19. β-ACTIN was used as a loading control, *n* = 4. **l** Sm-FISH of *Akr1b3* in *Shp2*^f/f and *Shp2*^d/d mice on day 19 with immunofluorescence of E-cadherin. **m** Pregnant length after injection of PBS (*n* = 12) and PGF2α (*n* = 12, 50 μg/mouse) intraperitoneally in *Shp2*^d/d mice on day 19. Data are shown as boxplots where midlines indicate medians, boxes indicate interquartile range and whiskers indicate minimum/maximum range. **n** The ratio of survival pregnant females after injection of PBS (*n* = 10) and PGF2α (*n* = 12, 50 μg/mouse) intraperitoneally in *Shp2*^d/d mice on day 19. Source data of **b**, **c**, **d**, **e**, **g**, **i**, **k** and **n** are provided as a Source Data file.

luminal epithelium on day 16 was primarily marked by steroid hormone biosynthesis genes, including *Srd5a1* (Steroid 5 alpha-reductase 1) and *Hsd11b1* (Hydroxysteroid 11-beta dehydrogenase 1). We also observed that *Akr1b7* (Aldo-keto reductase 1b7), one of the enzymes responsible for PGF2α synthesis cooperated with COX2, was also specifically expressed in luminal epithelium (Fig. 2g, h). The gland was mainly marked by the pathways of salivary secretion and *Prkg1* (Protein kinase cGMP-dependent 1) mediated cGMP-PKG signaling pathway. Previous investigations indicated that glands were important source of nutrients for both human and mouse[13, 14], while the unique feature of glands at later stage approaching delivery was rarely recognized. In this study, we identified some specifically expressed transcription factors for epithelium and gland, respectively, including well recognized *Msx1* (Msh Homeobox 1) and *Sox9* (SRY-box transcription factor 9), as well as two new transcription factors, *Ehf* (ETS homologous factor) and *Foxp1* (Forkhead box P1) (Fig. 2i). In addition, we also found some specific markers expressed in glands, such as *Krt23* (Keratin 23), *Gpx2* (Glutathione peroxidase 2), and *Slc1a1* (Solute carrier family 1 member 1), along with some previously known markers *Prss28* (Protease, serine 28), *Sox9* (SRY-box transcription factor 9) and *Prss29* (Protease, serine 29) (Fig. 2j and Supplementary Fig. 6c–d). These results charted both expressional and functional discrepancy of uterine luminal and glandular epithelium at later stage of pregnancy.

## The dynamic changes in the epithelium on days 16 and 19 of pregnancy

To dissect the dynamic changes of uterine epithelium on days 16 and 19 of pregnancy, we compared the gene expression and cell interactions at these two stages. We noticed that while cell types and composition were comparable at both stages, gene expression was significantly different (Fig. 3a). Functional analysis revealed that cellular senescence was highly enriched in day 16 columnar luminal epithelium, squamous luminal epithelium and glands, while the ErbB signaling pathway was highly enriched in day 19 columnar luminal epithelium and squamous luminal epithelium with significant enrichment of MAPK signaling pathway in day 19 columnar luminal epithelium (Fig. 3b). The cell-cell communication analysis revealed that although stromal cells (Str_2) dominated interactions with other cells, the cross-talk of luminal epithelium (LE_0) with other cell types was obviously changed between these two different days (Fig. 3c and Supplementary Fig. 8). Notably, HB-EGF-EGFR/ERBB4 signaling pathway appeared activated to mediate the interaction between epithelium and other cell types (Fig. 3d), which was further evidenced by the co-immunostaining of KI67 with CK8 on days 16 and 19 (Fig. 3e). Interestingly, detailed analysis revealed that *Hbegf* (Heparin binding EGF like growth factor) was highly expressed in LE_0, Gland_10 and prolifEpi_20 (Fig. 3f). For the receptors of *Hbegf*, *Egfr* (Epidermal growth factor receptor) was expressed in both epithelium, stroma and other cell types, while *Erbb4* was specifically expressed in LE_0 and

prolifEpi_20 (Fig. 3g). More importantly, the expression of *Hbegf* in LE_0 and Gland_10 was overtly increased on day 19, heralding the potential role of HB-EGF-EGFR/ERBB4 signaling pathway in epithelium proliferation and regeneration before parturition (Fig. 3h, i).

After the growth factors binding to their respective receptors, SHP2 is activated by binding to phosphotyrosine-containing receptors to incite RAS- ERK1/2 pathway which in turn promotes cellular proliferation and regulates other signaling processes[19, 25]. To figure out the significance of HB-EGF-EGFR/ERBB4 signaling pathway in epithelium reformation, SHP2 deficient uterine epithelial organoid derived from day 19 was applied. The incapable formation of SHP2 knockout organoid compared with WT (Fig. 3j) suggested the essential role of HB-EGF-EGFR/ERBB4 signaling pathway in the proliferation of epithelium.

## The changes of cell composition and gene expression in the absence of epithelial *Shp2* uterus

To globally delineate the cell composition changes and differentially expressed genes in different cell types at the absence of SHP2, the whole uteri after removal of decidua, placenta and fetus from *Shp2*^f/f and *Shp2*^d/d were subjected to scRNA-Seq (Fig. 4a, b). Loss of epithelial *Shp2* showed comparable cell types composition, but the number of glandular epithelium appeared diminished (Fig. 4c), which was further supported by fewer FOXA2 positive cells in *Shp2*^d/d decidua (Fig. 4d). Co-immunostaining of CK8 and E-cadherin and HE staining showed that the structure of luminal epithelium was grossly comparable in both genotypes (Fig. 4e and Supplementary Fig. 9). However, *Spp1* (Secreted phosphoprotein 1), a marker of luminal epithelium, upregulated in *Shp2*^d/d mice (Fig. 4f). Functional annotation analysis revealed that the genes in terms of ferroptosis and steroid synthesis were mainly enriched in *Shp2*^d/d luminal epithelium (Fig. 4g). Among these genes, aldo-keto reductase family 1 member B7 (*Akr1b7*) was validated by single melocular fluorescence in situ hybridization (sm-FISH) (Fig. 4h, i). Altogether, these results showed that epithelium *Shp2* loss contributed to the dysfunction of luminal epithelium.

## *Shp2*-deficient uteri show dysregulation in decidua before labor

Regarding the critical role of maternal decidua in coordinating interactions between the mother and feto-placental unit[26], we assessed the development of the decidua by detecting the expression of decidual marker *Prl8a2* and found that the localization was obviously decreased, which suggested epithelial *Shp2* deficiency was detrimental to the development of decidua (Fig. 5a). Interestingly, we also found the thickness of decidua was thinner in *Shp2*^d/d uteri by co-immunostaining PR (Progesterone receptor) and CK8 (Fig. 5b). To further elucidate the effects of epithelial *Shp2* on stromal cells, we analyzed the differentially expressed genes (DEGs) in different stromal cells (Fig. 5c, d). Signaling pathway enrichment analysis of these DEGs in different stromal cells revealed that genes associated with steroid biosynthesis were down-regulated and genes associated with PPAR

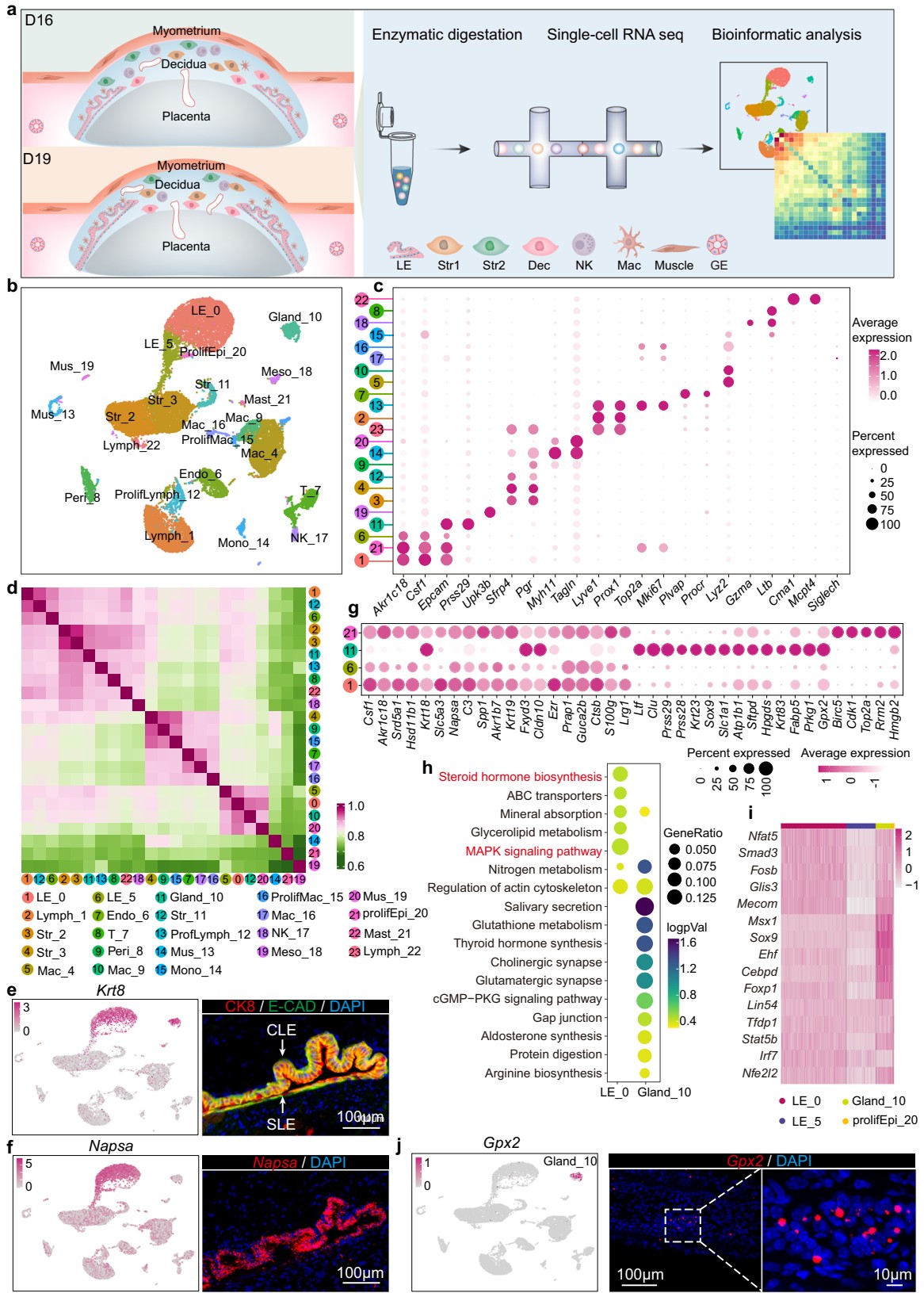

signaling pathway, glycolysis and nucleotide metabolism were up-regulated in *Shp2*$^{d/d}$ stromal cells (Fig. 5e). Fatty acid binding protein 4 (*Fabp4*), an important downstream target of PPAR signaling pathway which was highly expressed in decidua during early pregnancy[27,28], was specifically expressed in Str_2 and Str_9 and was significantly increased in *Shp2*$^{d/d}$ stromal cells (Fig. 5f–h). The above results suggested that epithelial SHP2 might participate in stroma development via epithelium-stromal interaction.

## The interactions between epithelium and other cell types

To thoroughly uncover the interactions between epithelium and other cell types, CellChat was utilized to probe the prioritized ligands in days

**Fig. 2 | The Single-cell landscape of mouse uteri and the characterization of epithelial cells. a** Diagram of the experimental and analysis workflow for single-cell transcriptome profiling in days 16 and 19 mouse uteri. LE luminal epithelial cells, Str1 stromal cells 1, Str2 stromal cells 2, Dec decidual cells, NK natural killer cells, Mac macrophage cells, Muscle muscle cells, GE glandular epithelial cells. **b** UMAP of the major cell types in days 16 and 19 uteri. Dots indicate individual cells and colors indicate different cell clusters. **c** Dot plot of marker genes in different cell types. Color bar represents average expression of genes. **d** The Pearson correlation coefficient of gene expression in each cell type. The numbers of colored rounds represent different cell types. Color bar represents correlation coefficient of different cell types. **e** UMAP visualization of the expression of *Krt8* and coimmunostaining of CK8 and E-CAD in day 19 uteri. CLE columnar luminal epithelium, SLE squamous luminal epithelium. This result was repeated independently in three individual mice with similar results. Color bar represents expression of indicated gene in cells. **f** UMAP visualization of the expression of *Napsa* and the location of *Napsa* in day 19 uteri via sm-FISH. This result was repeated independently in three individual mice with similar results. Color bar represents expression of indicated gene in cells. **g** Dot plot of marker genes for different epithelial cells. Color bar represents normalized expression of genes. **h** KEGG enrichment analysis of highly expressed genes in luminal and glandular epithelium. Dot size: number of genes in data attributed to each KEGG term. Color bar: the log10 transformation of enrichment pvalue. Significance based on over-representation test in clusterProfiler analysis with Benjamini-Hochberg-adjusted *p* values. **i** Heat map of transcription factors expression in four different epithelial cells. Color bar represents normalized expression of genes. **j** UMAP visualization of the expression of *Gpx2* and the location of *Gpx2* in day 19 uteri via sm-FISH. This result was repeated independently in three individual mice with similar results. Color bar represents expression of indicated gene in cells.

16 and 19 epithelial cells. During the investigation of the communications between epithelium and stroma, *Ifne* (Interferon epsilon), *Apoe* (Apolipoprotein E) and other factors were the primary factors to mediate these interactions. Especially, *Ifne*, a cytokine specifically expressed in uterine epithelium to protect from viral infection[29], was also highly expressed in epithelium by regulating *Ctsb* (Cathepsin B) and *Hsd11b2* (Hydroxysteroid 11-beta dehydrogenase 2) expression in stroma via interacting with *Ifnra1* (Interferon Alpha and Beta receptor subunit 1) and *Ifnra2* (Interferon Alpha and Beta receptor subunit 2) (Fig. 6a–e). The increased expression of *Ifne* in day 19 epithelium suggested a potential protective role of *Ifne* after parturition. Since there were plenty of macrophages around epithelium at later stage (Supplementary Fig. 10), we also investigated the cell-cell interactions between these two cell types. There were several prioritized cytokines in epithelium mediating the cross-talk between epithelium and macrophages, including *Il1a* (Interleukin 1 alpha), *Il33 (Interleukin 33)*, *Spp1*, *Csf1* and *Csf3* (Colony stimulating factor 3) (Fig. 6f–h). Among them, *Csf1*, highly expressed in pre-implantation uterus, had been reported as critical for macrophage differentiation and homing[30]. This high expression of *Csf1* and *Csf3* in epithelium would be the latent explanation for the macrophage distribution close to epithelium and is probably involved in endometrium repairing after parturition. In addition, we also identified the differentially expressed ligand-receptor pairs in the absence of epithelial SHP2, including decreased HB-EGF-ERBB4 and increased CFS1-CSF1R (Fig. 6i). Collectively, our studies depicted the previously unrecognized interactions dominated by epithelium (Fig. 7).

## Discussion

Parturition initiation involves the coordination of progressively orchestrated processes in decidua, placenta and fetus. Previous studies have demonstrated that PGF2α[21,31] and OXTR[32] play essential roles in parturition. The rapid development of scRNA-Seq shifts the focuses on the cell specific roles of different cell types at single cell resolution. Although the roles of other cell types in parturition have been reported more or less recently, the physiological significance of epithelial cells during the onset of labor is largely ignored. In vitro studies suggest epithelium-specific sodium channel (ENaC) promotes PGE2 production via COX2 in uterine epithelial cells through phosphorylation of CREB in the preparation of parturition[33]. But the regulatory mechanism of uterine epithelial cells at the onset of labor has not been illustrated for decades due to the lack of suitable genetic mouse models. In our study, we observe that uterine epithelial *Shp2* is pivotal in regulating local prostaglandin synthesis and epithelial-stromal interactions before parturition.

There is a dynamic reorganization of epithelium during pregnancy. The pre-implantation uterine epithelium mainly responds to embryo implantation through interacting with embryo and stromal cells[34]. After implantation, the epithelium around embryo disappears via entosis and apoptosis in order to facilitate the extensive communication between the embryo and stromal cells[35,36]. The remaining luminal epithelium at inter-implantation sites and glandular epithelium secret arrays of nutrient substances to nourish the developing fetus[15,37–40]. From day 10 of pregnancy, the epithelium undergoes regeneration to wrap the fetus[15]. While the mechanism of this epithelial regeneration is largely unknown. Furthermore, the physiological significance of epithelium in parturition preparation and intiation at late stage of pregnancy remains elusive. Our investigation not only provides information for this epithelium repairment but also an unappreciated mechanism for the niche maintenance by interacting with stromal cells and macrophages. Whether there are interactions with other cell types via factors secreted by epithelium deserves further investigation.

Prostaglandins, such as PGF2α derived from maternal decidua or fetal membrane, play significant roles in parturition via G protein-coupled receptors which transmit diverse signaling pathways, including PKC/Ca²⁺, Raf/MEK/ERK and Rho signaling pathways, to myometrium in order to ignite contraction and inflammation in decidua and myometrium[41–44]. It has been demonstrated that the prostaglandin activity in human amniotic epithelium is also important in initiating labor at term[45]. Our results prove that, apart from the aforementioned sites, epithelium is another critical site for PGs generation and parturition initiation. There is a significant decrease of COX1 and COX2 in epithelium on day 19 caused by epithelial *Shp2* deficiency which ultimately disturbs the synthesis of PGF2α. Previous studies have proposed that PR inhibits NF-κB activation and COX2 expression[46], whether *Shp2* function through these signaling pathways needs to be further investigated.

It is known that intricate and sophisticated conversations between epithelium and stroma play a pivotal role in uterine development and early pregnancy. Stromal cells are critical for the differentiation of the Mullerian epithelium at development stage via heterotypic recombinants of stroma and epithelium of uteri and vagina respectively[47]. IHH (Indian hedgehog signaling molecule), which is specifically expressed in uterine epithelium under the regulation of PR, GATA2 and SOX17[48,49], is a key factor in initiating epithelial-stromal communication network to regulate uterine stromal cell proliferation, embryo invasion and decidualization in both human and mice[50,51]. Stroma-derived IGF1 activates epithelial STAT3 via its epithelium specific receptor IGFR1 to regulate epithelial depolarization to ensure embryo implantation[52]. However, the role of epithelial-stromal interaction in the onset of labor remained elusive. Here, we show evidence that the disruption of crosstalk between epithelium and stroma, mediated by SHP2, impedes the development of stromal cells before labor. Further efforts are needed to figure out the underlying molecular mechanism. Moreover, considering the prescence of luminal and glandular epithelium in later stage uteri of prenancy, the development of luminal and glandular specific Cre would be of potential interest to dissect out cell specific roles these two cell types in parturition.

Taken together, this work reveals the significance of epithelium in parturition preparation and initiation by directing epithelial PGF2α

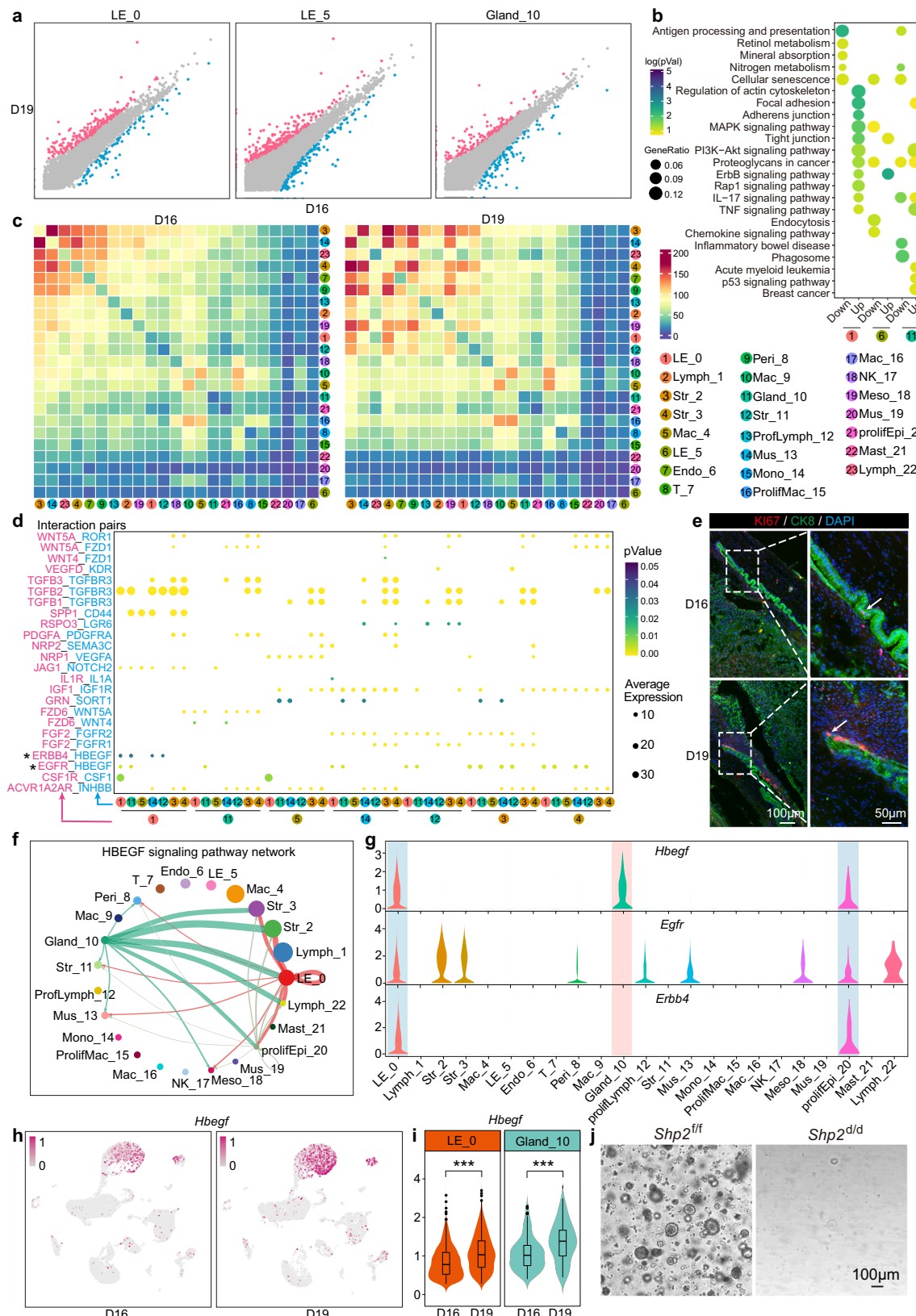

synthesis and maintaining microenvironment homeostasis dominated by epithelium. Moreover, our study uncovers the important role of SHP2 in epithelium regeneration. In conclusion, our results generate valuable insight regarding the underestimated role of uterine epithelium in parturition and could enlighten the exploration of potential strategic approach to targeting SHP2 to intervene PTB.

## Methods

### Animals and treatments

Two-month-old female mice were used in the present study. *Shp2*flox/flox mice with C57BL/6 background were constructed as previously described[53]. Uterine-epithelial-specific knock out mice were generated by crossing *Shp2*flox/flox mice with *Ltf*Cre/+ mice (a gift from Sudhansu K

**Fig. 3 | The difference of epithelial cells between days 16 and 19 uteri. a** Scatter plot depicting differentially expressed genes in LE_0, LE_5 and Gland_10 on days 16 and 19. Red and blue dots represent genes upregulated and downregulated in day 19, respectively. Gray dots represent genes with no obviously change between days 16 and 19. **b** KEGG enrichment of upregulated and downregulated DEGs in LE_0, LE_5 and Gland_10. Dot size: number of genes in data attributed to each KEGG term. Color bar: the log10 transformation of enrichment *P*-value. Significance based on over-representation test in clusterProfiler analysis with Benjamini-Hochberg-adjusted *p* values. Color bar represents log transformed *p*Value. **c** Abundance of connections between different cell types in days 16 and 19 uteri using CellphoneDB. Color bar represents the number of interactions between different cell types. **d** Dot plot illustrating detailed interaction pairs in different cell types. The size of dots represent the expression of ligands and receptors in corresponding cell types. Color bar represents the significance level. The numbers of colored rounds represent different cell types. *P* values are derived from one-sided permutation tests based on statistical framework in CellphoneDB. Color bar represents the interaction *p*Value of ligand and receptor. **e** Co-immunostaining of CK8 and KI67 in days 16 and 19 uteri. White arrows indicate KI67 positive cells. **f** Circle plot showing the supposed HBEGF signal pathway network among different cell types in day 16 uteri. **g** Violin plot showing expression of ligand-receptor pairs of HBEGF signal pathway among each cell type in day 16 uteri. **h** UMAP visualization of the expression of *Hbegf* in days 16 and 19 epithelial cells of uteri. Color bar represents expression of indicated gene in cells. **i** Violin plot depicting quantitative expression of *Hbegf* in LE_0 and Gland_10 between days 16 and 19. *p* = 2.2e-16 in LE_0 and *p* = 1.5e−7 in Gland_10 between days 16 and 19, one-sided Wilcoxon tests. Data are shown as boxplots where midlines indicate medians, boxes indicate interquartile range and whiskers indicate minimum/maximum range. **j** Organoid growth of *Shp2*^f/f and *Shp2*^d/d epithelial cell from day 19 uteri. This result was repeated independently in three individual mice with similar results.

Dey, Cincinnati Children's Hospital, Cincinnati, Ohio, USA)[54]. Mice were housed in the laboratory animal center of Xiamen University, in accordance with the guidelines for the care and use of laboratory animals. All experimental procedures were approved by the Animal Welfare Committee of Research Organization (X200811), Xiamen University. Female mice at least two-month-old were mated with fertile wild-type males to induce pregnancy (vaginal plug = day 1 of pregnancy). Litter size and the date of delivery were monitored throughout the duration of the pregnancy. Implantation sites were visualized by intravenous injection of 100 μl Chicago blue dye in saline on day 5 of pregnancy. Uterine horns were flushed to check for the presence of embryos if no blue band was observed. PGF2α (50 μg/mouse, Cayman) or PBS was administered intraperitoneally on day 19 of pregnancy to induce parturition in epithelial SHP2 deleted mice. Parturition and litter size were monitored by daily observation of mice in the morning, noon and evening. Delayed delivery was defined as birth occurring later than day 20 of pregnancy. At least three independent mice in each genotype were used for each individual experiment.

## Immunostaining
Immunohistochemistry was performed in 10% neutral buffered formalin-fixed paraffin-embedded sections or frozen sections. For paraffin sections, after deparaffinized, the slides were incubated in citrate buffer for antigen retrieval by hyperbaric heating and then incubated overnight at 4 °C with the primary antibodies including COX2 (Abcam,1:200). A Histostain-SP Kit (Zhongshan Golden Bridge Biotechnology) was applied to visualize the antigen. For immuno-fluorescence staining, 4% formaldehyde-fixed frozen sections were blocked in 5% Bovine serum albumin (BSA) in PBST (0.1% Triton X-100) and then incubated with primary antibodies including SHP2 (Santa Cruz, 1:200), CK8 (DSHB, 1:500), FOXA2 (Abcam, 1:500), COX2 (Abcam, 1:200), PR (CST, 1:200), F4/80 (CST, 1:200), ERα (Santa Cruz Biotechnology, 1:500), KI67 (Servicebio, 1:200) and E-cadherin (CST, 1:200). Specific secondary antibodies labeled with Cy3 or Cy2 were utilized. The images were captured using a Leica DM2500 light microscope and Zeiss LSM 880+Airyscan. All antibodies used in our studies are listed in Supplementary Table 1.

## In situ hybridization
In situ hybridization was modified according to that previously described[55]. Frozen sections were mounted onto poly-L-lysine-coated slides and fixed in 4% paraformaldehyde in PBS for 1 h at room temperature. Then the slides were hybridized to specific Digoxingenin (DIG)-labeled cRNA probes overnight at 55 °C. After hybridization, the slides were blocked in blocking buffer and incubated with Alkaline Phosphatase conjugated anti-Digoxingenin antibody and visualized by NBT/BCIP substrate. The images were captured using a Leica DM2500 light microscope. Mouse-specific anti-sense cRNA probe for *Akr1c18* can be found in Supplementary Table 2.

## Sm-FISH
Single-molecule fluorescence in situ hybridization (sm-FISH) was applied based on a previously established SCRINSHOT protocol[56,57]. First, frozen sections were fixed in 4% PFA for 1 h and blocked in blocking buffer with oligo dT (0.1 μM, sangon) and tRNA (0.2 μg/μl, Roche). Then slides were hybridized with padlock probes designed via the PrimerQuest online tool (Integrated DNA Technologies: IDT), which included a denaturation step at 55 °C for 15 min and an annealing step at 45 °C for 2 h, followed by ligation of the padlock probes overnight at 25 °C using SplintR ligase (0.5 U/μl, NEB). Next, the cyclic probes on sections began on rolling circle amplification (RCA) at 30 °C using a RCA primer (0.1 μM, sangon) with thiophosphate modification and φ29-polymerase (0.5 U/μl, Vazyme). After RCA, a fixation with 4% PFA for 15 min was done to stabilize the RCA products on the sections. Finally, the RCA products were incubated with a detection probe conjugated with HRP (0.04 μM, sangon) and visualized by tyramide signal amplification (TSA, 1:500, Servicebio) according to the manufacturer's instructions. The images were captured using Zeiss LSM 880+Airyscan. Padlock probes used are listed in Supplementary Table 3 with blue representing the complementary sequence to mRNA and the other representing the constant sequence of the padlock probes.

## Culture of mouse uterine epithelial organoid
Mouse uteri from day 19 of pregnancy in *Shp2*^f/f and *Shp2*^d/d mice were harvested and cut into several pieces, then incubated with a digestion solution buffer containing 6 mg/ml dispase (Gibco) in HBSS at 37 °C for 40 min. The whole epithelium was collected through squeezing uteri with a flattened needle and incubated in 6 mg/ml dispase and 25 mg/ml pancreatin (Sigma) at 37 °C for 30 min. The digested cells were passed through a 70 μm filter to obtain single epithelial cells. Cells were suspended with phenol red-free Dulbecco modified Eagle medium and Ham F12 nutrient mixture (Gibco) (DMEM/F12, 1:1) and then mixed with Matrigel (Corning) on ice at a ratio of 1:6. The mixture was immediately plated onto 24-well plates and organoid medium was added after Matrigel freezing. The organoid medium was reported as previously stated[58].

## Measurement of serum E2, P4, and uterine PGF2α levels
Mouse blood and uteri samples were respectively collected on day 19 for serum E2, P4, and PGF2α levels by radioimmunoassay (RIA) at the Fifth People's Hospital of Xiamen University. The values are expressed as mean ± SEM.

## Western blot
Samples were homogenized in RIPA buffer with protease and phosphatase inhibitors and quantified using a BCA kit. The protein extracts were run on a 10% Bis-acrylamide gel and transferred to PVDF membranes. After blocking in 5% non-fat milk, the membranes were

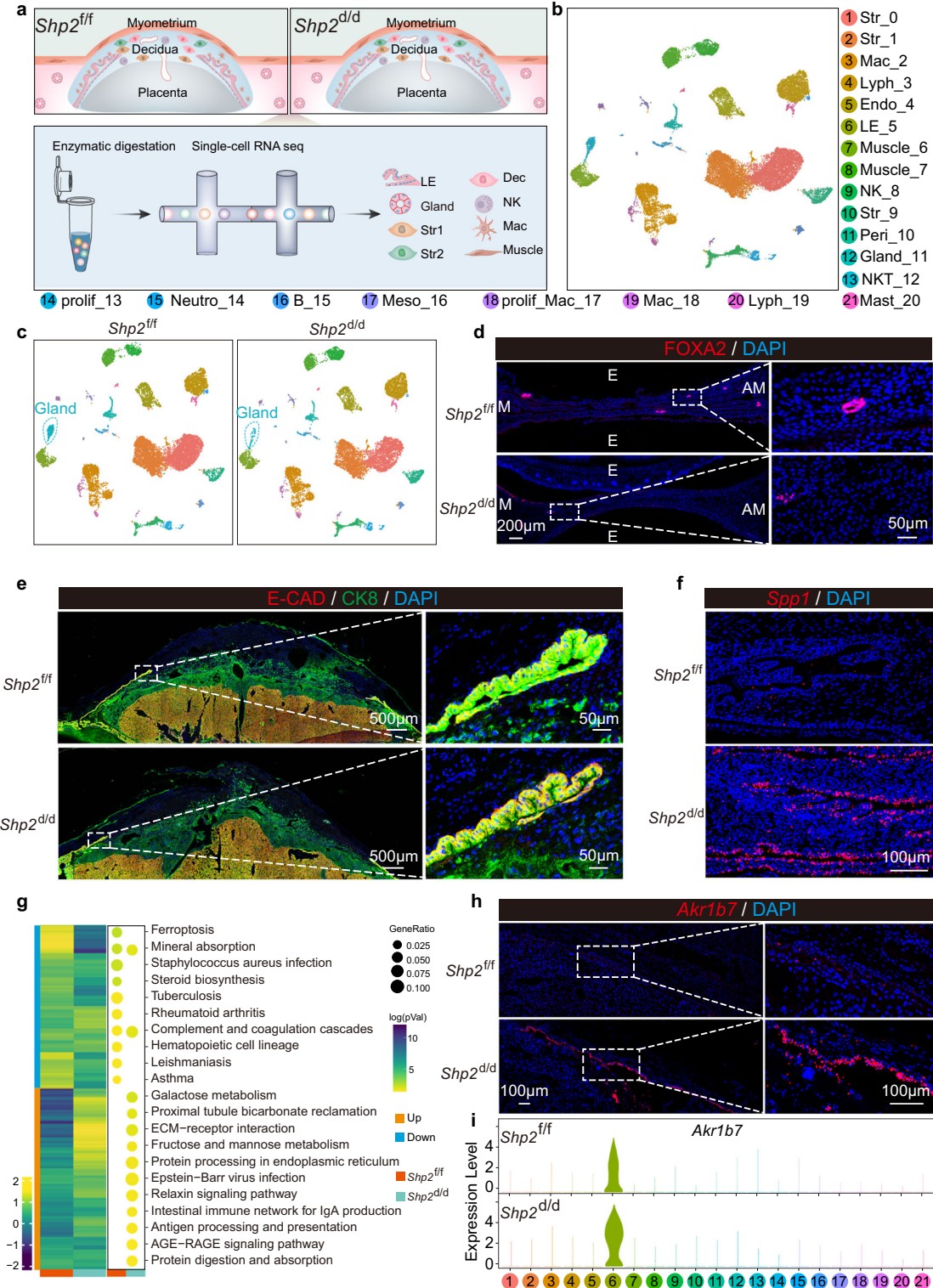

**Fig. 4 | Epithelium changes due to uterine epithelial *Shp2* loss. a** Diagram of the experimental workflow for single-cell transcriptome profiling in *Shp2*[f/f] and *Shp2*[d/d] mouse uteri on day 19. **b** UMAP of the major cell types in *Shp2*[f/f] and *Shp2*[d/d] mouse uteri on day 19. **c** UMAP visualization of different cell types in *Shp2*[f/f] and *Shp2*[d/d] mouse uteri on day 19. **d** Immunofluorescence of FOXA2 in inter-implantation sites of *Shp2*[f/f] and *Shp2*[d/d] mice on day 19. This result was repeated independently in three individual mice with similar results. **e** Coimmunostaining of E-cadherin and CK8 in *Shp2*[f/f] and *Shp2*[d/d] mouse uteri on day 19. This result was repeated independently in three individual mice with similar results. **f** Location of *Spp1* mRNA in *Shp2*[f/f] and *Shp2*[d/d] mice epithelial cell via sm-FISH. This result was repeated

independently in three individual mice with similar results. **g** Heatmap and KEGG functional analysis of differentially expressed genes in LE_5 for *Shp2*[f/f] and *Shp2*[d/d] mouse uteri on day 19 with color bar representing z-score normalized expression of genes and the log10 transformation of enrichment *P*-value, respectively. Dot size: number of genes in data attributed to each KEGG term. Significance based on over-representation test in clusterProfiler analysis with Benjamini-Hochberg-adjusted *P* values. Color bar represents log transformed *p*Value. **h** Location of *Akr1b7* mRNA in *Shp2*[f/f] and *Shp2*[d/d] mice epithelial cell via sm-FISH. This result was repeated independently in three individual mice with similar results. **i** Violin plot showing expression of *Akr1b7* in *Shp2*[f/f] and *Shp2*[d/d] mice on day 19.

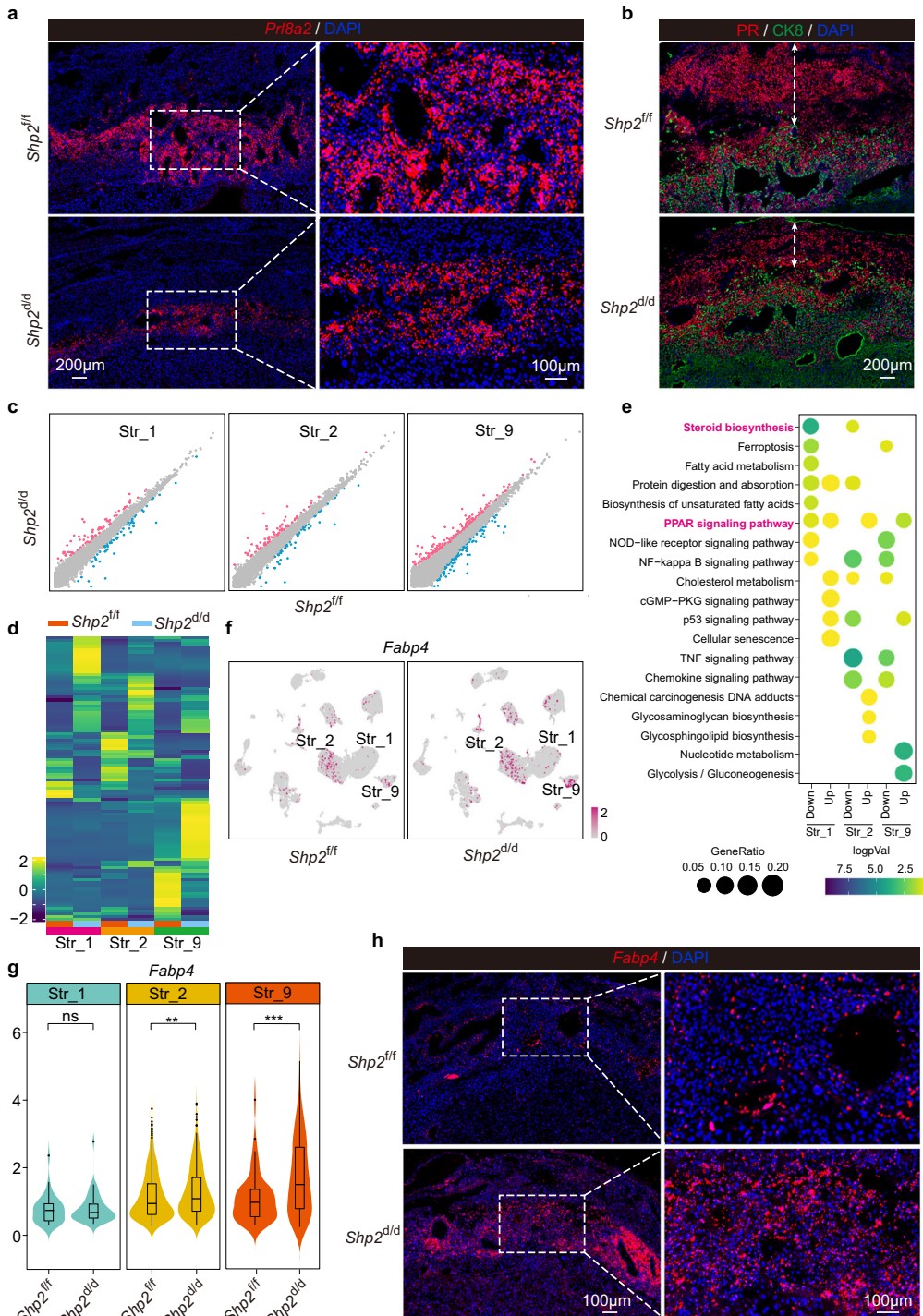

**Fig. 5 | Epithelial *Shp2*-deficient uteri displayed abnormal in decidua. a** Sm-FISH of *Prl8a2* in *Shp2*^f/f and *Shp2*^d/d mice decidua on day 19. This result was repeated independently in three individual mice with similar results. **b** Co-immunofluorescence of PR and CK8 in decidua of *Shp2*^f/f and *Shp2*^d/d mice on day 19. The dash line with double arrow indicates the thickness of decidua. This result was repeated independently in three individual mice with similar results. **c** Scatter plot depicting differentially expressed genes in Str_1, Str_2 and Str_9 of *Shp2*^f/f and *Shp2*^d/d mice on day 19. Red and blue dots represent genes upregulated and downregulated in *Shp2*^d/d mice, respectively. Gray dots represent genes with no obvious change between *Shp2*^f/f and *Shp2*^d/d mice. **d** Heatmap of differentially expressed genes in Str_1, Str_2 and Str_9 for *Shp2*^f/f and *Shp2*^d/d mouse uteri on day 19. Color bar represents z-score normalized expression of genes. Color bar represents normalized expression of genes. **e** KEGG functional analysis for DEGs in

Str_1, Str_2, and Str_9 of *Shp2*^f/f and *Shp2*^d/d mice on day 19. Dot size: number of genes in data attributed to each KEGG term. Color bar: the log10 transformation of enrichment *P*-value (logpVal). Significance based on over-representation test in clusterProfiler analysis with Benjamini-Hochberg-adjusted *P* values. Color bar represents log transformed *p*Value. **f** UMAP visualization of the expression of *Fabp4* in *Shp2*^f/f and *Shp2*^d/d mice stromal cells. Color bar represents expression of indicated gene in cells. **g** Quantitative expression of *Fabp4* in Str_1, Str_2 and Str_9 between *Shp2*^f/f and *Shp2*^d/d mice. \*\**p* = 0.0011, \*\*\**p* = 0.00016, one-sided Wilcoxon tests. Data are shown as boxplots where midlines indicate medians, boxes indicate interquartile range and whiskers indicate minimum/maximum range. **h** Location of *Fabp4* mRNA in *Shp2*^f/f and *Shp2*^d/d mice decidua via sm-FISH. This result was repeated independently in three individual mice with similar results.

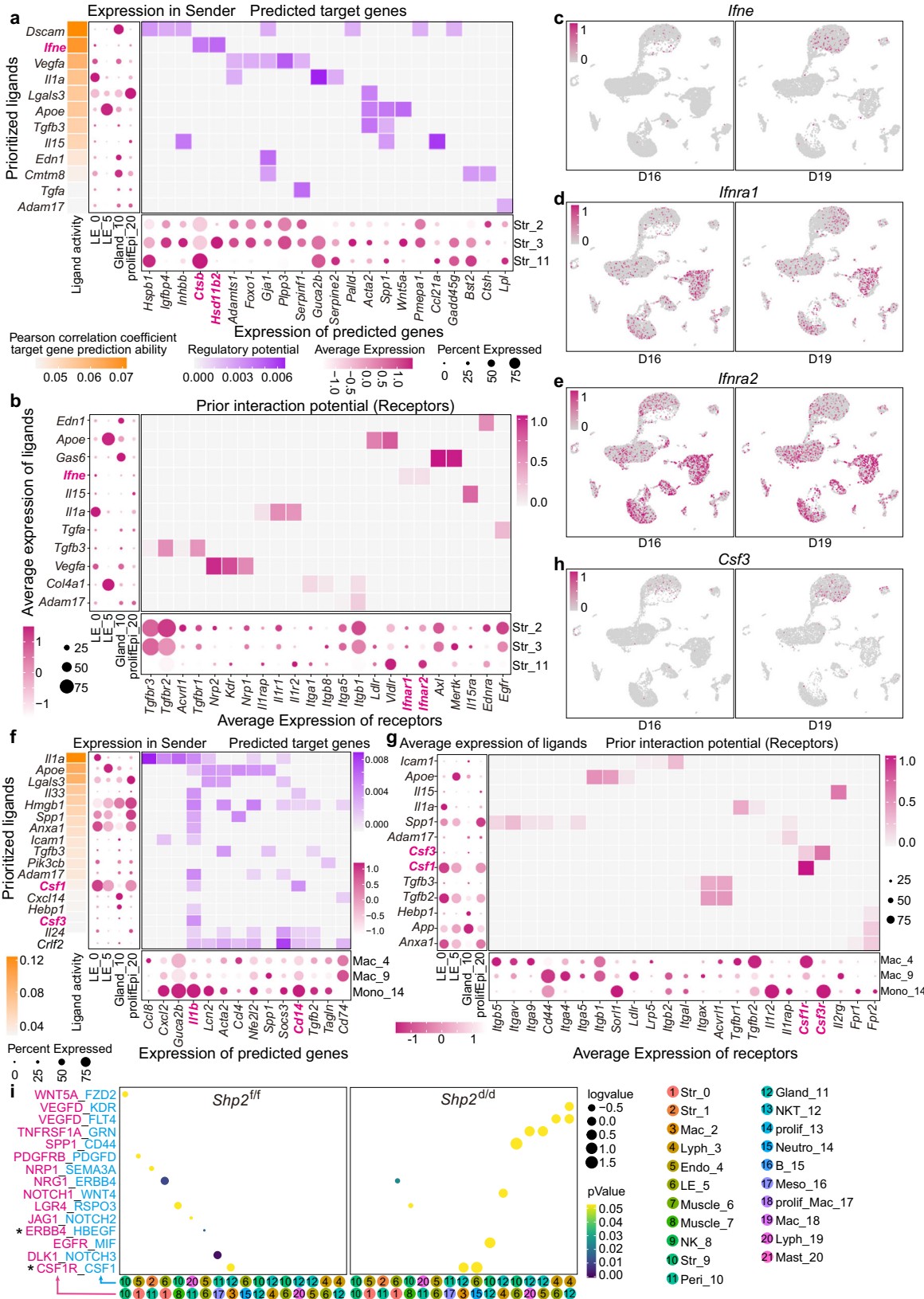

**Nature Communications** | (2023)14:7356

incubated overnight at 4 °C with primary antibodies including STAR (Abcam, 1:1000), COX1 (Abcam, 1:1000) and β-Actin (CST, 1:3000). The membranes were then washed in TBST and incubated with HRP conjugated secondary antibodies. The bands were visualized with an ECL kit. Bands were visualized using ChemiDocTMXRS+ (BIO-RAD) according to the manufacturer's instructions.

**Single-cell RNA sequencing of mouse uteri**

Uteri from three individual on days 16 and 19 of pregnancy in *Shp2*^f/f mice or day 19 of pregnancy in *Shp2*^f/f and *Shp2*^d/d mice were harvested. After removing the fetus, placenta and part of the decidua, the uteri were minced into small pieces, and then incubated with digestion solution containing collagenase II (160 U/

**Fig. 6 | Cell-cell interactions between epithelial cells and other cell types. a** The interactions between epithelial cells (LE_0, LE_5, Gland_10 and prolifEpi_20) and three stromal cells (Str_2, Str_3 and Str_11). The most left represents ligands of epithelium defined by NicheNet Pearson correlation and they are ranged by the ability of each ligand to their targets. The dot plots indicate their different expression of ligands in epithelial cells and predicted targets in stromal cells. Heatmap represents the predicted ligands activity by NicheNet on their target genes in stromal cells. Yellow color bar represents the Pearson correlation coefficient, purple color bar represents regulatory potential and magenta color bar represents nomalized expression of genes. **b** Heatmap represents the interactions between ligands in epithelial cells (LE_0, LE_5, Gland_10 and prolifEpi_20) and their receptors in stromal cells (Str_2, Str_3 and Str_11). Dot plots represent the average expression levels of ligands and receptors in epithelial and stromal cells, respectively. Color bar in the bottom left corner represents nomalized expression of genes, color bar in the top right corner represents interaction potential. **c**–**e** UMAP visualization of the expression of *Ifne, Ifnra1* and *Ifnra2* in days 16 and 19 mice. Color bar represents expression of indicated gene in cells. **f** The interactions between epithelial cells (LE_0, LE_5, Gland_10, and prolifEpi_20) and macrophages (Mac_4, Mac_9, and Mono_14). The most left represents ligands of epithelium defined by NicheNet Pearson correlation and they are ranged by the ability of each ligand to

their target. The dot plots indicate their different expression of ligands in epithelial cells and predicted targets in macrophages and monocytes. Heatmap represents the predicted ligands activity by NicheNet on their target genes in macrophages. Yellow color bar represents the Pearson correlation coefficient, purple color bar represents regulatory potential and magenta color bar represents nomalized expression of genes. **g** Heatmap represents the real interactions between ligands in epithelial cells (LE_0, LE_5, Gland_10 and prolifEpi_20) and targets in macrophages (Mac_4, Mac_9 and Mono_14). Dot plots represent the average expression levels of ligands and receptors in epithelial and macrophages and monocytes, respectively. Color bar in the bottom left corner represents nomalized expression of genes, color bar in the top right corner represents interaction potential. **h** UMAP visualization of the expression of *Csf3* in days 16 and 19 epithelium. Color bar represents expression of indicated gene in cells. **i** The changed ligand-receptor pairs between different cell types in *Shp2*^f/f and *Shp2*^d/d mice determined by CellphoneDB. The size of dot: the expression of ligands (pink) and receptors (blue) in different cell types, color bar: the significance of enriching. The numbers of colored round representing different cell types. *P* values are derived from one-sided permutation tests based on statistical framework in CellphoneDB. Color bar represents *p*Value of ligand receptor.

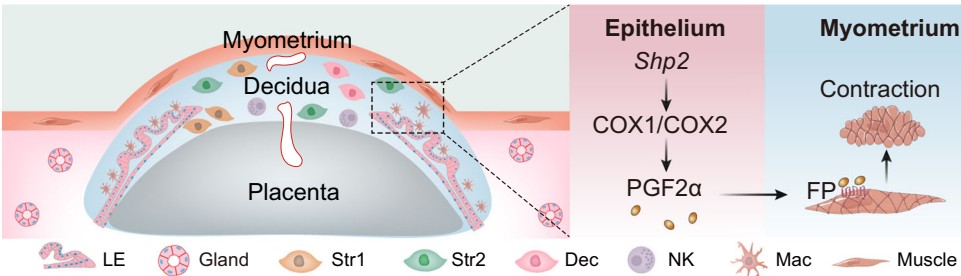

**Fig. 7 | A representative scheme of a physiological role of epithelial SHP2 in parturition initiation.** SHP2 incites myometrium contraction by regulating PGF2α production in epithlium to participate smooth muscle contraction via justacrine. LE

Luminal epithelial cells, Str1 stromal cells 1, Str2 stromal cells 2, Dec decidual cells, NK natural killer cells, Mac macrophage cells, Muscle muscle cells, Gland glandular epithelial cells.

ml, Worthington) and DNase I (7.5 µg/ml, Roche) at 37 °C for 60 min with agitation. The solution was passed through a 40 µm cell strainer. After centrifugation at $400 \times g$ for 5 min, the cell pellet was incubated with 1 mL of RBC lysis buffer on ice for 5 min. The dead cells were removed using a Dead cell remove kit (Miltenyi) following the manufacturer's protocol, and cell number and viability were analyzed using 0.4% trypan blue. After generating single cell gel beads in the emulsion (GEM) according to the manufacturer's protocol (10X Genomics, CG000183), a single-cell library was generated using the Chromium Single Cell 3' Reagent Kits (10X Genomics). The library was sequenced on the Illumina HiSeq X Ten system.

### Single-cell RNA sequencing data analysis

Reads were processed using the CellRanger 4.0.0 pipeline with the default and recommended parameters. The output of CellRanger was then imported into the Seurat (v3.0) R toolkit for quality control and downstream analysis. All functions were run with default parameters unless specified otherwise. Low-quality cells (total UMI count per cell (library size) below 30,000, genes per cell <500 and the content of mitochondria genes > 20%) were excluded. Next, we used a cluster-level approach to remove potential doublet cells. In brief, the doublet score was calculated for each cell using the doubletCells function of R package v.1.18.7. Cell clusters in each sample were identified by examining the top 50 principal components (PCs) across highly variable genes (HVGs), building a neighbor graph by the buildSNNGraph function, and then clustering using the cluster_louvain function from the igraph R package v.1.2.9. The median doublet score of each cell cluster was calculated using a median-centred MAD-variance normal

distribution. Clusters with a median score above the extreme top end of this distribution were considered doublets. After filtering, the remaining cells were kept for downstream analysis.

### Cell clustering and annotation

The HVGs were selected using the highly_variable_genes function in Seurat. Nearest neighborhood graphs were built using the neighbors function, and the community algorithm was applied for clustering using the louvain function. The markers of characterized cell types in our single cell RNA sequencing were confirmed by FindAllMarkers. The major cells types identified in our dataset were annotated based on well-known marker genes.

### Ligand-receptor interactions analysis by CellPhoneDB

To access cellular crosstalk between different cell types in the control and experimental group, ligand-receptor analysis was performed using the CellphoneDB repository, a public database that stores receptors, ligands, and their interactions[8]. Both ligands and receptors included subunit structures that accurately represented heteromeric complexes in order to infer the cell-cell communication networks from single-cell transcriptome data. In addition, the specific interactions between indicated cell types were generated by NicheNet (v.1.1.0) to interrogate possible interactions and target genes between indicated cell types.

### Cell chat

To further identify and compare cell-cell interactions, we applied CellChat, an open-source R package (https://github.com/sqjin/CellChat), to analyze our single cell sequencing data. Following

the standard process, including analysis and identification of OverExpressedGenes, identifyOverExpressedInteractions, and projectData with an official parameter set, we calculated the potential intercellular communications among the epithelium and other cell clusters.

## Statistical analysis

All data are shown as the mean ± SEM. Unless otherwise noted, each experiment was repeated three or more times with biologically independent samples. GraphPad Prism (v.9.0.0) and R (v.4.1.0) were used for experimental statistical analysis. Comparisons between groups were conducted using Student's *t*-tests, Wilcoxon rank-sum tests, Fisher's exact test and ANOVA. *P* value less than 0.05 were considered to be statistically significant.

## Reporting summary

Further information on research design is available in the Nature Portfolio Reporting Summary linked to this article.

## Data availability

Raw data generated in this study have been deposited at the National Center for Biotechnology Information Sequence Read Archive under accession number PRJNA936400 [https://www.ncbi.nlm.nih.gov/sra?term=SRP423274]. Source data are provided with this paper.

## Code availability

Codes used in this study have been deposited at Zenodo [https://doi.org/10.5281/zenodo.10022180].

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

## Acknowledgements

This work was supported by the National Key R&D Program of China (2022YFC2704500 and 2022YFC2704600 to W.D.; 2022YFC2704500 to Z.L.; 2021YFC2700302 to H.W.; 2022YFC2702400 to J.L.), Basic Science Center Program of the National Natural Science Foundation of China (82288102 to H.W.), National Natural Science Foundation of China (82271720 and 31970797 to Z.L.; 82288102 to H.W.; 82122026, 32171117 and 81971419 to W.D.; 82222026 and 32270907 to S.K.; 31971071 to J.L.).

## Author contributions

M.L., M.J., J.C., Y.L., YD.T., YP.T., H.Z., Y.W., S.Z., L.Z., X.L., and X.X. performed experiments and prepared figures. W.D., G.F., H.B., S.K., J.L., and Z.L. designed experiments. W.D., M.L., and M.J. analyzed data. W.D., M.L., Z.L., and H.W. wrote the manuscript.

## Competing interests

The authors declare no competing interests.
