## [Peer Review File · Nature Communications]

REVIEWER COMMENTS

Reviewer #1 (Remarks to the Author):

Manuscript ID: NCOMMS-23-09237

Title: Deciphering an indispensable role of uterine epithelial SHP2 in parturition initiation at single cell resolution

Authors: Liu et al.

This manuscript by Liu et al describes a potential role for the non-receptor protein tyrosine phosphatase Shp2 in parturition initiation. The manuscript entails multiple aspects starting with single cell analyses of uteri excluding the conceptus at day 16 and 19 of pregnancy, a functional assessment of uterine epithelial-specific ablation of Shp2, the early role of Shp2 in contributing to normal decidualization, a potential role for SHP2 in inflammatory signals, and uterine-stromal interactions that are mostly important in early pregnancy. The authors report delayed onset of parturition as a result of uterine epithelial-specific Shp2 KO, elevated circulating progesterone compared to control, and higher prevalence of neonatal death. Loss of Shp2 also contributed to reduced number of endometrial glandular epithelial cells and dysfunction of luminal epithelial cells. Finally, the manuscript reports reduced immune response, and decidual dysregulation in Shp2-cKO mice

Parts of the data are interesting and novel, specifically the role of uterine epithelial Shp2 on parturition timing. However, as a whole the study appears very disjointed. The rationale for the focus on Shp2 as a result of identifying "EGFR/MAPK signaling" enriched in the epithelial compartment of scRNA-seq data is ill-justified. The subsequent analyses jump between an early role for Shp2 in uterine epithelium and epithelial-stromal interactions to that in late-gestation for the initiation of parturition. Analyses on cell-cell interactions and inflammatory signal mediators lack in depth. Overall, this study contains some interesting results that, after restructuring, would be better suited for a more specialized journal.

Major comments:

The sole rationale for conducting an in-depth analysis of Shp2 seems to hinge on the enrichment of HB-EGF/EGFR signaling in the luminal compared to the glandular epithelium. There is little justification for the resultant choice of Shp2. Moreover, the conditional knockout strategy applied does not distinguish between luminal and glandular epithelium, so this does not make sense.

As stated by the authors, the re-formation of the uterine epithelial lining at, or preceding, parturition remains poorly understood. As this is the major focus of this study, a detailed histological analysis of implantation sites including their surrounding myometrial tissue should be conducted and included in the main part to depict when and how the epithelium reforms. It is stated that this process is initiated at

the inter-implantation sites, which then should harbour leading edges of highly proliferative epithelial cells. This must be demonstrated. It should also be shown if any glandular epithelium remains at E19 in the thin muscular tissue layer overlying the decidua. What is the squamous epithelium, and where does it come from (e.g., Fig. 1e)?

The text clearly states that the tissue for single cell analysis entailed the uterus after removal of decidua, placenta and fetus. It is confusing that the diagrams also include decidua. The myometrial tissue (i.e. the tissue overlying the decidual face of implantation sites) at E19 is a very thin layer. There are barely any glands in this thin strip of muscle. Even at E16, as shown in the supplement, there is basically no gland in the tissues analysed (the picture only shows one small gland at the periphery which appears to have been chosen for the main figure). Hence it is unclear where all of the epithelial cells, specifically the glandular epithelial cells, should come from that were identified in the single cell clusters. Again, a histological depiction of the entire structure at low magnification and zoomed in areas of interest are instrumental to gain a proper understanding of tissue morphology and morphogenetic processes around parturition.

As an example, in Fig. 3a, does this really depict the uterus overlying the decidua, or rather an inter-implantation site? It is surprising that there should be such extensive stretches of epithelial cells present in the tissue covering the implantation site.

In line with this question around the proportional representation of cells in the scRNA-seq analyses, the proportion of epithelial cells in Fig. 4d and Fig. 1b is highly discrepant, Fig. 4 is more in keeping with expectation as for cell proportions. What is the difference?

Also, what is the cytokeratin-positive compact zone which, hence, is of fetal origin?

The Ltf-Cre model induces gene deletion in the uterine epithelium at E4. Many of the data shown argue in favor of a role for Shp2 in early epithelial-stromal interactions, that lead to later differences in the decidua, as shown. They do not, however, relate to the role of SHP2 in parturition. These developmentally dynamic roles of SHP2 remain unresolved and inconclusive.

It is unclear how the few glands overlying the decidua at E19 should cause such major differences in PGF2a production that result in far-ranging consequences on systemic maternal hormone levels. Also, there is no obvious spatial relationship between the PGF2a-production by glands and the changes in COX1 and COX2 expression. How can this be explained?

Leading on from this question, how tight is the Ltf-Cre induced conditional ablation of Shp2? Does it affect the squamous epithelial layer shown in Fig. 1e? Are other estrogen-responsive epithelia outside the uterus affected?

The lack of epithelial organoid growth from uteri at E4 does not relate to a role for Shp2 in parturition.

There are multiple occasions of mix-ups in the figures vs legends between E16 and E19 that confuse the results.

There is no decidualization at labor. This process is completed far earlier. This statement in the text (line 367) encapsulates the major pitfalls of this manuscript that mixes up processes that occur at very different developmental time points.

The link to TLR4 mediated inflammatory responses is poorly developed and perhaps unnecessary for a manuscript focussed on Shp2 function in the induction of labor.

How does uterine epithelial-specific ablation of Shp2 affect the ovary? This link is poorly evolved.

The manuscript confuses insights from the mouse model with those in humans. Clear distinctions should be made in all statements that relate to either one or the other. For example, the term 'uterine milk' is usually only applied in humans. Similarly, 'prostaglandin activity in amniotic epithelium as an important initiator of labor at term' refers to the situation in humans.

The manuscript needs major contextual editing for language.

Additional points:

The manuscript requires major improvements on the following points:

1. The inconsistent use of "mouse uteri" throughout the text is distracting. Please, substitute "mice uteri" and "mice uterus" to "mouse uteri".
2. Suppl Fig 2A – this figure would benefit from higher magnification and a double staining for epithelial cell markers alongside ERa. It is currently unclear what part of the tissue one is looking at, so clarification in the figure legend and appropriate labelling is required.
3. Fig 1E, 1F, and 1J – all showing D16 (as compared to Sup Fig 2B-C) but figure legend says D19. Please revise as this is key information from your results and it is crucial to be reporting the correct timing.
4. Line 458: Please provide full organoid expansion medium composition with final concentration of reagents in a table in Supplementary information. Boretto et al (2017) (reference 61) report varying

concentrations of Rspodin-1 and Wnt3A as a test in their publications, hence this current manuscript should indicate what has been used.

5. Line 174: Justification needed of why authors are using day 4 and day 19 Shp2-cKO mice

6. Line 249: Unsure where this fits in this research – why focus on inflammation now and why jump from delayed onset of labor to inflammation induced PTB therapy?

Reviewer #2 (Remarks to the Author):

The manuscript by Liu and coworkers describes the endometrial cell types and impact of SHP2 in the endometrial epithelium in the non labor (day 16) and labor (day 19) mouse uterus. This manuscript gives important data on the cell types and potential communication pathways in the mouse uterus during this period. It also shows transcriptomic changes in these cell types at these stages. The manuscript then goes to investigate the role of SHP2 in the endometrial epithelium by conditional ablation of this gene using the LtfCre model. This analysis shows that loss of SHP2 in the epithelium results in a delay in parturition. The delay is in part due to a decrease in prostaglandin synthesis and this delay can be rescued by Prostaglandin treatment.. This is an important manuscript because it not only defines the endometrial cell types during pregnancy and labor but also shows a critical role for the endometrial epithelium during this process. There are two minor weaknesses in this manuscript

1. The LPS experiment is not well developed. It states that LPS was given on Day 4. Why was this day chosen and not later in pregnancy. What is the significance of the RNA seq in this approach since the actions of SHP2 are later in pregnancy. This data should be removed.

2. What is the mechanism of SPH2 regulation of parturition. IN the discussion it is suggested that it may involve the regulation of P4 signaling. This should be evalusted at least bioinformatically by comparing the gene expression changes to know genes identified by transcriptomics or cistromics to PGR signaling.

All this is an important and novel finding and will add significant new avenues for research.

Reviewer #3 (Remarks to the Author):

General

The manuscript authored by M. Liu and associates describes experiments to establish the factors relevant to the differences in gene expression between the late gestation and the periparturient uteri in the mouse. The methods employed include scRNAseq and consequent bioinformatic analysis, ChatCell to determine the cell interactions in the uterus around parturition and conditional depletion of a gene, Shp2 in the uterine epithelium. The results demonstrate that deletion of this gene does not affect embryo implantation or establishment of the deciduum, but that it has an effect on the late pregnancy changes in this tissue that normally accompany the birth of the litter. This is a novel and noteworthy finding, as it opens a new horizon in exploration of parturition. A strength of the manuscript is the cogent discussion of this new and important information. In terms of the methods employed, it would be useful to have more information of the quality control aspects of the scRNA analysis.

This manuscript has much to recommend it. It is a novel exploration resulting in the discovery of a previously unknown mechanism, i.e. the role of the uterine epithelium in the process of parturition. The single cell global gene analysis appears to have been appropriately conducted and the presence of scatterplots in the figures is indicative of the high quality of the data. The manuscript will require extensive language editing, as there are many, many syntax, grammar and spelling errors that render it difficult to read and detract from the overall presentation.

Specific

1. Line 32: It would seem that decidualization, a process that begins at implantation on day 5 of gestation was normal, not aberrant. The process that was disturbed was parturition in this study, thus the abstract is misleading. This requires clarification.
2. Line 119: It is stated that the signature of epithelial cells in labor is portrayed. Although parturition in the mouse usually occurs on day 19 post mating, there is some variability. How was it determined that the animals were in labor at the time of collection of the uterine tissues?
3. Figure 1F is not labelled.
4. Figure 3C: There is no information about whether these differences are statistically significant.
5. Lines 188 et seq: Was it determined whether Shp2 was depleted in the ovary?
6. Figure 3F: The immunohistochemical image purported to indicate the overexpression of StAR is not particularly convincing.
7. Figure 3F: There is a significant reduction in the expression of Akr1c18 in the corpora lutea of the d/d model attributed to reduced expression of Ptgs2. Is it possible that Akr1c18 is a direct target of Shp2?
8. Figure 7 (summary figure) is not an easily understandable resumé of the investigation. It should be improved, or a better and more comprehensive figure legend be provided.

My co-authors join me in expressing our sincere appreciation to the reviewers' thoughtful comments. We have critically reviewed each comment and added new experiments and results as suggested by the reviewers. Our responses are elaborated below. The reviewers' comments are followed by our responses highlighted in blue. All changes in the revised manuscript are marked by blue.

Reviewer #1

Parts of the data are interesting and novel, specifically the role of uterine epithelial Shp2 on parturition timing. However, as a whole the study appears very disjointed. The rationale for the focus on Shp2 as a result of identifying "EGFR/MAPK signaling" enriched in the epithelial compartment of scRNA-seq data is ill-justified. The subsequent analyses jump between an early role for Shp2 in uterine epithelium and epithelial-stromal interactions to that in late-gestation for the initiation of parturition. Analyses on cell-cell interactions and inflammatory signal mediators lack in depth. Overall, this study contains some interesting results that, after restructuring, would be better suited for a more specialized journal.

The sole rationale for conducting an in-depth analysis of Shp2 seems to hinge on the enrichment of HB-EGF/EGFR signaling in the luminal compared to the glandular epithelium. There is little justification for the resultant choice of Shp2. Moreover, the conditional knockout strategy applied does not distinguish between luminal and glandular epithelium, so this does not make sense.

Thanks very much for this suggestive comment. There are four reasons for conducting the in-depth analysis of *Shp2*: (1) MAPK signaling pathway is enriched in epithelium; (2) this enrichment is increased in day 19 uterus compared with day 16; (3) there was low organoid formation efficiency derived from *Shp2* deficient epithelial cells. (4) SHP2 is a critical nonreceptor protein tyrosine phosphatase to activate Ras/Mitogen-activated protein kinase (MAPK) pathway (PMID: 17993263). Hence, we constructed a genetic mouse model harboring uterine epithelium specific deletion of *Shp2* by crossing *Shp2-loxp* mice (*Shp2^{fl/fl}*) with *Ltf-Cre* mice to further dissect out the physiological role of SHP2 in both luminal and glandular epithelium in parturition (Lines 173-179).

Currently, there is no luminal or glandular epithelium specific Cre available yet due to limited understanding between these two types of epithelia. Although FOXA2 is highly expressed in glands, *Foxa2*-driven Cre recombinase express in node, notochord, floorplate, and endoderm as well as endoderm-derived organs including lung, liver, pancreas, and gastrointestinal tract throughout development (PMID: 18798232), which largely limit the application of this Cre in female reproductive tract. In additional, our scRNA-Seq result unravel several glandular specific genes, it might be possible to dissect out the roles of different epithelium in parturition initiation by creating new tool mice driven by these genes, which deserve further investigation. Additionally, our results indicate that the receptors of growth factor only expressed in luminal epithelium but not glands, the application of *Ltf-Cre* is feasible to illustrate the role of *Shp2* in epithelium cells. We revised the manuscript

to avoid the disjointedness between HB-EGF/EGFR signaling and SHP2 in epithelium. The limitation of LTF-Cre to distinguish luminal and glandular epithelium is also discussed (Lines 366-370).

To illustrate the role of *Shp2* in organoid formation the late gestation, we separated epithelial cell on day 19 to detect the efficiency of organoid formation. The result suggested that organoid formation is obviously disrupted from day 19 in the absence of epithelial SHP2 (Lines 179-182).

Fig. 1 The role of SHP2 for epithelial growth in the late gestation. Organoid growth of *Shp2^{f/f}* and *Shp2^{d/d}* epithelial cell from day 19 uteri.

As stated by the authors, the re-formation of the uterine epithelial lining at, or preceding, parturition remains poorly understood. As this is the major focus of this study, a detailed histological analysis of implantation sites including their surrounding myometrial tissue should be conducted and included in the main part to depict when and how the epithelium reforms. It is stated that this process is initiated at the inter-implantation sites, which then should harbour leading edges of highly proliferative epithelial cells. This must be demonstrated. It should also be shown if any glandular epithelium remains at E19 in the thin muscular tissue layer overlying the decidua. What is the squamous epithelium, and where does it come from (e.g., Fig. 1e)?

Thanks for this concern very much. The observation of epithelium reformation based on histological analysis during pregnancy has been reported before (Fig. 2, PMID: 6624690, ref 15, Line 73). The epithelium reformation is started around day 10 at AM site, then at both M and AM sites (indicated by red arrows). Our immunostaining of CK8 in days 4,8 and 10 also confirms the distribution of epithelium (Fig. 3, 4).

To detect the presence of proliferation epithelium during pregnancy, KI67 was co-stained with CK8 in days 12 and 14 uterus. The results show that KI67 positive cells mainly locate at the end of leading edges as marked by white dashed box (Fig. 4). Based on this

result, it is inappropriate to state that “the epithelium undergoes regeneration from inter-implantation sites”. This sentence is changed to “From day 10 of pregnancy, the epithelium undergoes regeneration to wrap the fetus” (Lines 330-331)

To detect the presence of glandular epithelium at day 19, we performed HE and FOXA2 staining in day 19 uterus. The result show that there is FOXA2 positive glandular epithelium in inter-implantation site (Fig. 5). Based on previous studies (PMID: 31074826; PMID: 29426931), glands mainly distribute in the M site of uterus, which is largely different from human. Our previous work based on tissue clearing and 3D imaging uncover the distribution and dynamic change of glands during early pregnancy (Fig.5-8; PMID: 29426931). The results also show that glands are mainly existed in AM site. The epithelium remains at E19 in the thin muscular tissue layer overlying the decidua is primary luminal epithelium.

It is a very good question about the origination of the squamous epithelium. There are few functional studies about the role of squamous epithelium during pregnancy. Previous study supposes that the epithelium covering decidualized stromal cells (close to fetus) is squamous epithelium, while the epithelium covering non-decidualized stromal cells is columnar epithelium (PMID: 6624690). The underlying molecular mechanism regulating the development of these two different cell types deserve further study, which is out of the scope of current study.

Fig. 2 Illustration of dynamic change of epithelium during pregnancy from days 6 to 15(PMID: 6624690). The M: mesometrial;AM antimesometrial; Em: embryo. Red arrow: epithelium.

Fig. 3 The histological analysis of implantation sites at days 4, 8 and 10. The immunostaining of CK8 in day 4, 8, and 10 uteri. M: mesometrial; AM: antimesometrial; E: embryo; T: trophoblast cells; Epi: epithelium cells.

Fig. 4 There are proliferative epithelial cells as re-formation of the uterine epithelium. The coimmunostaining of CK8 and KI67 in days 12 and 14 uteri. E: embryo; P: placenta.

Fig. 5 The location of glandular epithelium at day 19 in inter-implantation sites. HE staining and FOXA2 immunostaining in inter-implantation sites of day 19 uteri. M: mesometrial; AM: antimesometrial; E: embryo.

The text clearly states that the tissue for single cell analysis entailed the uterus after removal of decidua, placenta and fetus. It is confusing that the diagrams also include decidua.

The myometrial tissue (i.e. the tissue overlying the decidual face of implantation sites) at E19 is a very thin layer. There are barely any glands in this thin strip of muscle. Even at

E16, as shown in the supplement, there is basically no gland in the tissues analysed (the picture only shows one small gland at the periphery which appears to have been chosen for the main figure). Hence it is unclear where all of the epithelial cells, specifically the glandular epithelial cells, should come from that were identified in the single cell clusters. Again, a histological depiction of the entire structure at low magnification and zoomed in areas of interest are instrumental to gain a proper understanding of tissue morphology and morphogenetic processes around parturition.

Stroma cells (including decidualized and undecidualized stromal cells) are the most abundant cell type in the maternal decidua. To increase the enrichment of epithelium, stromal cells are removed manually as much as possible, but it is difficult to remove stromal cells thoroughly due to the tightly connection between stromal and other cell types.

As illustrated by our previously work applying whole uterine staining, tissue clearing and two-photo microscope imaging, the glands are evenly distributed in the uterus before embryo implantation, while after embryo implantation, the glands in the implantation site are extended and pushed out due to the rapid growth of decidua and embryo on day 6 and day 8 (PMID: 29426931). With regard to the glands in the inter-implantation site, they distribute crowd ascribed to the growing embryo of both sides (Fig. 6-8). Our histology result also showed that there was array of FOXA2 positive glands in inter-implantation site on day 19 (Fig. 5). At the implantation site, the areas of interest are the leading edge covering the placenta and the other end connect with inter-implantation site (Fig. 9).

Fig. 6 3D images of day 5 uteri. Images of one uterine horn in *Rosa26^{tdTomato}Ltf^{Cre/+}* mice on day 5. IS: Implantation site, inter-IS: interimplantation site; M: mesometrial; AM: antimesometrial; * indicates embryos. (PMID: 29426931)

Fig. 7 3D images of day 6 uteri. 3D immunostaining with CDH1, segmented and 3D rendered images in day 6 implantation sites. asterisk indicates embryos; arrows indicate the glands in the inter-implantation site. (PMID: 29426931)

Fig. 8 3D images of day 8 uteri. 3D immunostaining with CDH1, segmented and 3D rendered images in day 8 implantation sites. asterisk indicates embryos; Arrows show the sheared ducts from the gland lobules. (PMID: 29426931)

Fig 9. The histological depiction of the entire structure. The coimmunostaining of CK8 and E-cadherin in day 19 uteri. The upper dashed rectangle boxes represent epithelium cover decidualized stromal cells and placenta, the lower dashed rectangle boxes represent epithelium in inter-implantation site.

As an example, in Fig. 3a, does this really depict the uterus overlying the decidua, or rather an inter-implantation site? It is surprising that there should be such extensive stretches of epithelial cells present in the tissue covering the implantation site.

As illustrated above (Fig. 5 and 9), the leading edge of the epithelium at implantation site cover the decidua and placenta, while the epithelium in inter-implantation is very different. The upper white dashed box of Fig. 9 represents the area in Fig. 3A in the manuscript.

In line with this question around the proportional representation of cells in the scRNA-seq analyses, the proportion of epithelial cells in Fig. 4d and Fig. 1b is highly discrepant, Fig. 4 is more in keeping with expectation as for cell proportions. What is the difference?

Thanks for this concern very much. The tissue collection of these two scRNA-Seq datasets is different. Fig. 1b mainly collect tissue from days 16 and 19, while Fig.4d collect tissue from day 19 in WT and Shp2KO mice. These two batch scRNA-Seq were carried out separately, the process of tissue collection, especially the remaining of stromal cells in placenta separation, digestion and others might contribute to this cell proportion discrepancy.

Also, what is the cytokeratin-positive compact zone which, hence, is of fetal origin?

It is a very good question. At the later stage of pregnancy, the growing TGCs invade into maternal tissue. The compact zone contains both PR positive maternal tissue and cytokeratin-positive fetal tissue characterizing with high-density DAPI staining due to the squeezing of maternal tissue (Figure. 10).

Fig. 10 The histological depiction of decidua. The immunostaining of PR in day 19 uteri. CZ: compacting zone; JZ: junctional zone; LZ: labyrinth zone.

The Ltf-Cre model induces gene deletion in the uterine epithelium at E4. Many of the data shown argue in favor of a role for Shp2 in early epithelial-stromal interactions, that lead to later differences in the decidua, as shown. They do not, however, relate to the role of SHP2

in parturition. These developmentally dynamic roles of SHP2 remain unresolved and inconclusive.

Thanks for this concern. The epithelial-stromal interactions were mainly investigated in days 16 and day 19 based on scRNA-Seq. The data in early stage of pregnancy (day 4) provide evidence that embryo implantation and decidualization were normal in both genotypes, excluding the possibility of delayed parturition arise from the defect of embryo implantation or decidualization. The major topic of this study is to unravel the role of epithelium in parturition. We first depict that there is extensive interaction of epithelium and other cell types approaching parturition. Then, our study further uncovers the underlying mechanism of epithelium in parturition via SHP2. In a word, this study proves evidence that the developmentally dynamic roles of SHP2 in delayed parturition at the absence of epithelial SHP2 is limited. And the developmental role of epithelial SHP2 in other aspects apart from parturition deserves further investigation.

It is unclear how the few glands overlying the decidua at E19 should cause such major differences in PGF2a production that result in far-ranging consequences on systemic maternal hormone levels. Also, there is no obvious spatial relationship between the PGF2a-production by glands and the changes in COX1 and COX2 expression. How can this be explained?

Thanks for this constructive concern. It has been well established that decidual COX2 derived PGF2 α is critical for luteolysis through circulation of uterus and ovary (PMID: 9751758) and then contribute to the change of systemic hormone levels. Our data showed that the critical enzymes for PGF2 α synthesis, such as *Ptgs1* and *Akr1b3*, expressed in luminal epithelial cells at implantation site and luminal and glandular epithelium at inter-implantation sites by co-staining with CK8 (Fig. 11A-C). These results suggest that, apart from stroma cells in decidua, epithelium, including both luminal and glandular epithelium, is also another important site for PGF2 α generation. The down-regulated *Ptgs1* and *Akr1b3* in *Shp2* deficient epithelium indicate the important role of SHP2 in epithelial PGF2 α -production (Line 226-228).

Fig 11. The spatial relationship between the PGF2 α -production and COX1 and COX2 expression in *Shp2^{fl/fl}* and *Shp2^{d/d}* mice on day 19.

A Sm-FISH of *Akr1b3* by co-staining with CK8 in inter-implantation sites of *Shp2^{fl/fl}* and *Shp2^{d/d}* mice on day 19. M: mesometrial; AM: anti-mesometrial; E: embryos.

B Sm-FISH of *Ptgs1* by co-staining with CK8 in inter-implantation sites of *Shp2^{fl/fl}* and *Shp2^{d/d}* mice on day 19. M: mesometrial; AM: anti-mesometrial; E: embryos.

C Sm-FISH of *Akr1b3* by co-staining with CK8 in implantation sites of *Shp2^{fl/fl}* and *Shp2^{d/d}* mice on day 19.

Leading on from this question, how tight is the Ltf-Cre induced conditional ablation of Shp2? Does it affect the squamous epithelial layer shown in Fig. 1e? Are other estrogen-responsive epithelia outside the uterus affected?

Thanks for this constructive suggestion. The epithelium specific deletion efficiency of SHP2 was provided in Fig. 3A in the manuscript. To estimate whether squamous epithelial layer was affected at the absence of SHP2, we conduct co-immunostaining of CK8 and E-cadherin and HE staining in *Shp2^{fl/fl}* and *Shp2^{d/d}* uterine (Fig. 12A-B). The result indicates that the squamous epithelial layer was not affected in *Shp2* deficient epithelium on day 19 (Line 249-250).

The original study of the establishment of Ltf-iCre mice show that there is little to low Cre recombinase in other estrogen-responsive epithelium, such as vagina and oviduct, but with strong Cre recombinase activation in uterine epithelium, seminal vesicle, cauda epididymis, ductus deferens and caput epididymis (PMID: 24823394). We also evaluate other estrogen-responsive epithelia outside the uterus, such as oviduct and cervix of day 19 in *Shp2^{fl/fl}* and *Shp2^{d/d}* mice. The histological structure is comparable in both genotypes (Fig. 13A-B).

Fig. 12 The structure of epithelium was not affected in the absence of Shp2.
A Coimmunostaining of E-cadherin and CK8 in *Shp2^{fl/fl}* and *Shp2^{d/d}* mouse uteri on day 19.
B HE staining in *Shp2^{fl/fl}* and *Shp2^{d/d}* mouse uteri on day 19.

Fig. 13 The epithelium of cervix and oviduct was not affected in *Ltf*-Cre induced conditional ablation of *Shp2*.

A, HE staining in *Shp2^{fl/fl}* and *Shp2^{d/d}* mouse cervix on day 19.

B, HE staining in *Shp2^{fl/fl}* and *Shp2^{d/d}* mouse oviduct on day 19.

The lack of epithelial organoid growth from uteri at E4 does not relate to a role for Shp2 in parturition.

Thanks for this concern very much. It is inappropriate to speculate the role of epithelial SHP2 in parturition from early pregnant stage. Organoid growth from day 19 epithelium has been also estimated. Our results suggest that SHP2 deficiency compromised epithelium growth and organized formation at later stage (Fig.1), which further corroborate the observation that sufficient epithelium regeneration is important for parturition (Line 181-182).

There are multiple occasions of mix-ups in the figures vs legends between E16 and E19 that confuse the results.

Thanks for this reminding very much. We had checked our manuscript carefully and modified these errors in text.

There is no decidualization at labor. This process is completed far earlier. This statement in the text (line 367) encapsulates the major pitfalls of this manuscript that mixes up processes that occur at very different developmental time points.

Thanks for this comment very much. It is true that the decidualization is initiated at early stage in both human and mice. While there is considerable decidualized stromal cell in maternal part in mice. The localization of decidualization marker *Pr18a2* indicate that these decidualized stromal cells mainly localize in the compact zone (Fig 5A in our revised manuscript). In our scRNA-Seq, since the decidualized stromal cells were removed manually, there is no decidualized stromal in our scRNA-seq datasets. The presence of decidualized stromal cells is proved by multiple scRNA-Seq and functional studies in mice approaching parturition (PMID: 36599348, 31067461, 27454290, 23979163). The exist of decidualized stromal cells in human has also been proven as there are also PRL positive cells at labor, although the number is significant less than mouse (PMID: 35260533).

The link to TLR4 mediated inflammatory responses is poorly developed and perhaps unnecessary for a manuscript focused on Shp2 function in the induction of labor.

Thanks for this suggestive concern. To exclude the distraction of major focus of this study of the physiological significance of epithelial SHP2 in parturition, the part of TLR4 mediated inflammatory responses is removed in the revised manuscript.

How does uterine epithelial-specific ablation of Shp2 affect the ovary? This link is poorly evolved.

Decidua derived PGF2 α is critical for luteolysis in ovary approaching parturition as stated in question 5 above. Previously study had proven that COX-1-deficient mice show delayed parturition due to impaired luteolysis accompanied with elevated serum progesterone concentration (PMID: 9751758). In this study, COX-1 expression in epithelium is significantly downregulated by SHP2, which in turn affects luteolysis and contributed to unsuccessful parturition initiation.

The manuscript confuses insights from the mouse model with those in humans. Clear distinctions should be made in all statements that relate to either one or the other. For example, the term 'uterine milk' is usually only applied in humans. Similarly, 'prostaglandin activity in amniotic epithelium as an important initiator of labor at term' refers to the situation in humans.

Thanks for this suggestion. We have carefully modified our manuscript to distinguish the statements of mouse models and these in humans. For example, we have changed the term 'uterine milk' to 'glands were important source of nutrients for both human and mouse' (Line 136-137), and modified 'prostaglandin activity in amniotic epithelium as an important initiator of labor at term' into 'prostaglandin activity in human amniotic epithelium as an important initiator of labor at term' (Line 344-345).

The manuscript needs major contextual editing for language.

We have made major language editing of this manuscript by a native speaker.

Additional points:

1. The inconsistent use of "mouse uteri" throughout the text is distracting. Please, substitute "mice uteri" and "mice uterus" to "mouse uteri".

We had substitute all "mice uteri" and "mice uterus" to "mouse uteri".

2. Suppl Fig 2A – this figure would benefit from higher magnification and a double staining for epithelial cell markers alongside ER α . It is currently unclear what part of the tissue one is looking at, so clarification in the figure legend and appropriate labelling is required.

We have conducted a double staining for epithelial cell marker, CK8, alongside ER α at decidua on day 19 (Fig. 14), and we have modified the figure legend and added corresponding label (Line 990-992).

Fig. 14 The expression of ER α in day 19 mouse uteri. Immunofluorescence staining of ER α in 19 mouse uteri. GE: columnar epithelium; SE: squamous epithelium.

3. Fig 1E, 1F, and 1J – all showing D16 (as compared to Sup Fig 2B-C) but figure legend says D19. Please revise as this is key information from your results and it is crucial to be reporting the correct timing.

Thanks for this reminding. The figure legend was verified in the revised new manuscript.

4. Line 458: Please provide full organoid expansion medium composition with final concentration of reagents in a table in Supplementary information. Boretto et al (2017) (reference 61) report varying concentrations of Rspodin-1 and Wnt3A as a test in their publications, hence this current manuscript should indicate what has been used.

Thanks for this suggestion, the composition and concentration of organoid expansion medium has been provided in revised manuscript and listed as below.

Supplementary Table 4 organoid medium composition

reagent	Cat	Final concentration
DMEM/F12	Gibco 11039-021	
penicillin/streptomycin	Gibco 15140-122	1%
ITS	Gibco 41400-045	1%
L-Glutamine	Gibco 21051-024	2 mM
Nicotinamide	Sigma N3376	1 mM
B27	Gibco 17504-044	2%
N2	Gibco 17502-048	1%
EGF	Peptotech 315-09	50 ng/ml
FGF-10	Peptotech 100-26	100 ng/ml
Noggin	Peptotech 250-38	100 ng/ml
WNT-3A	Peptotech 315-20	200 ng/ml
R-Spondin-1	Peptotech 120-38	200 ng/ml
A83-01	MCE HY-10432	0.5 μ M
Nac	Sigma A7250	1.25 mM
Y27632	WAKO	10 μ M

5. Line 174: Justification needed of why authors are using day 4 and day 19 Shp2-cKO mice.

The major phenotype of epithelial SHP2 deficient mice is delayed parturition. To exclude whether this phenotype is originated from defect of early stage of pregnancy, day 4, the day before embryo implantation and gland extension, is selected for histomorphological and deletion efficient analysis. Since there embryo implantation and decidualization appears comparable in both genotypes, day 19, the day approaching parturition with higher expression of contraction associated proteins, including OXTR and CX43, is selected to analyze the difference of cell heterogeneity and gene expression in different cell types.

6. Line 249: Unsure where this fits in this research – why focus on inflammation now and why jump from delayed onset of labor to inflammation induced PTB therapy?

Thanks for this suggestive reminding. To focus more on the major finding of this studying on the failure of parturition, the part about inflammation and inflammation induced PTB therapy have been removed in revised manuscript as stated above.

Reviewer #2

The manuscript by Liu and coworkers describes the endometrial cell types and impact of SHP2 in the endometrial epithelium in the non labor (day 16) and labor (day 19) mouse uterus. This manuscript gives important data on the cell types and potential communication pathways in the mouse uterus during this period. It also shows transcriptomic changes in these cell types at these stages. The manuscript then goes to investigate the role of SHP2 in the endometrial epithelium by conditional ablation of this gene using the LtfCre model. This analysis shows that loss of SHP2 in the epithelium results in a delay in parturition. The delay is in part due to a decrease in prostaglandin synthesis and this delay can be rescued by Prostaglandin treatment. This is an important manuscript because it not only defines the endometrial cell types during pregnancy and labor but also shows a critical role for the endometrial epithelium during this process. There are two minor weaknesses in this manuscript

1. The LPS experiment is not well developed. It states that LPS was given on Day 4. Why was this day chosen and not later in pregnancy. What is the significance of the RNA seq in this approach since the actions of SHP2 are later in pregnancy. This data should be removed.

Thanks for this suggestive information. After careful consideration and per the suggestion of reviewer 1 and 2, LPS experiments are removed in our revised manuscript due to the distraction of this part of experiments.

2. What is the mechanism of SPH2 regulation of parturition. IN the discussion it is suggested that it may involve the regulation of P4 signaling. This should be evalusted at least bioinformatically by comparing the gene expression changes to know genes identified by transcriptomics or cistromics to PGR signaling.

Thanks very much for this suggestive comment. In current study, our evidences suggest that SHP2 participates parturition by regulating epithelial COX1 expression and PGF2a production to facilitate luteolysis. To further evaluate the effect of SHP2 on PR signaling pathway, we compare the differentiated expressed genes (DEGs) in both stromal and epithelial cells with known genes identified by cistromics of PGR (GSM857546: PR ChIP-Seq of P4 treated uteri in ovariectomized mice; GSM5964410: PR ChIP-Seq in day 14). The results show that one third to half of DEGs in both epithelium and stroma are PR target genes in both datasets (Fig. 15). These results confirm that PR signaling is altered in epithelial SHP2 deficient mice.

Fig. 15 The overlapping of PR-target genes evidenced by PR ChIP-Seq with DEGs in both epithelium and stroma revealed by scRNA-Seq.

Reviewer #3

The manuscript authored by M. Liu and associates describes experiments to establish the factors relevant to the differences in gene expression between the late gestation and the periparturient uteri in the mouse. The methods employed include scRNAseq and consequent bioinformatic analysis, ChatCell to determine the cell interactions in the uterus around parturition and conditional depletion of a gene, Shp2 in the uterine epithelium. The results demonstrate that deletion of this gene does not affect embryo implantation or establishment of the deciduum, but that it has an effect on the late pregnancy changes in this tissue that normally accompany the birth of the litter. This is a novel and noteworthy finding, as it opens a new horizon in exploration of parturition. A strength of the manuscript is the cogent discussion of this new and important information. In terms of the methods employed, it would be useful to have more information of the quality control aspects of the scRNA analysis.

1. This manuscript has much to recommend it. It is a novel exploration resulting in the discovery of a previously unknown mechanism, i.e. the role of the uterine epithelium in the process of parturition. The single cell global gene analysis appears to have been appropriately conducted and the presence of scatterplots in the figures is indicative of the high quality of the data. The manuscript will require extensive language editing, as there are many, many syntax, grammar and spelling errors that render it difficult to read and detract from the overall presentation.

We have made extensive language editing to avoid syntax, grammar and spelling errors in the manuscript.

2. Line 32: It would seem that decidualization, a process that begins at implantation on day 5 of gestation was normal, not aberrant. The process that was disturbed was parturition in this study, thus the abstract is misleading. This requires clarification.

Thanks for this comment. We have modified the abstract to avoid the misleading of the abstract.

3. Line 119: It is stated that the signature of epithelial cells in labor is portrayed. Although parturition in the mouse usually occurs on day 19 post mating, there is some variability. How was it determined that the animals were in labor at the time of collection of the uterine tissues?

This is a very good suggestion. The uterus needs to be appropriately prepared before parturition characterized with increased expression contraction-associated proteins (CAP) at day 19, including OXTR and CX43. Then, the muscle layer will become contractive to expel the fetus at the night of day 19. In current study, the major scope is to dissect the significance of epithelial SHP2 during labor preparation. The statements that “the animals were in labor” were misleading, and had been change to “the animals were before labor”.

4. Figure 1F is not labelled.

The Figure 1F had been labelled.

5. Figure 3C: There is no information about whether these differences are statistically significant.

The statistical significance is calculated by Fisher’s exact test and the p value has been added in the revised manuscript.

6. Lines 188 et seq: Was it determined whether Shp2 was depleted in the ovary?

Ltf-driven Cre recombinase have shown no express in ovary based on previously work (Fig. 16) (PMID: 24823394). We also detected the expression level of *Shp2* in ovaries of both genotypes on day 19. The results showed that *Shp2* expression was not affected in *Shp2* deficient ovary (Fig. 17).

Fig 16. The expression of $Ltf^{Cre/+}$ in ovary. Conditional gene recombination induced by $Ltf-iCre$ was visualized by lacZ staining in adult ovary. cl, corpus luteum; fl, follicle; ic, interstitial cell. (PMID: 24823394)

Fig. 17 The mRNA levels of $Shp2$ in $Shp2^{fl/fl}$ and $Shp2^{d/d}$ ovary. Quantitative real-time PCR analysis of $Shp2$ in $Shp2^{fl/fl}$ and $Shp2^{d/d}$ ovary on day 19. The values are normalized to $Gapdh$ and indicated as the mean ± SEM (n=3 biologically independent samples). Two-tailed unpaired Student's t-test, ns not significant.

7. Figure 3F: The immunohistochemical image purported to indicate the overexpression of StAR is not particularly convincing.

Thanks very much for this suggestive information. To further confirm our ISH result of $Star$, we also did real-time PCR experiment to compare the expression of $Star$ in the ovaries of both genotypes. There is no significant change of $Star$ mRNA expression (Fig. 16). Although the mRNA level of $Star$ detected by ISH and real-time PCR is not changed very much, the protein levels of STAR is much higher in SHP2 deficient ovary which might be due to the translational modification of STAR as reported before (PMID: 19321517).

Fig. 16 The mRNA levels of *Star* in *Shp2^{fl/fl}* and *Shp2^{d/d}* ovary. Quantitative real-time PCR analysis of *Star* in *Shp2^{fl/fl}* and *Shp2^{d/d}* ovary on day 19. The values are normalized to *Gapdh* and indicated as the mean ± SEM (n=3 biologically independent samples). Two-tailed unpaired Student's t-test, ns: not statistically significant.

8. Figure 3F: There is a significant reduction in the expression of *Akr1c18* in the corpora lutea of the d/d model attributed to reduced expression of *Ptgs2*. Is it possible that *Akr1c18* is a direct target of *Shp2*?

Since *Ltf* does not express in the ovary (PMID: 24823394) and the mRNA level of *Shp2* does not change in *Shp2^{fl/fl}* and *Shp2^{d/d}* ovary, it is unlikely that *Akr1c18* is a direct target of *Shp2* in corpora lutea. There might be other mechanism underlying *Akr1c18* expression in *Shp2^{d/d}* ovary.

9. Figure 7 (summary figure) is not an easily understandable resumé of the investigation. It should be improved, or a better and more comprehensive figure legend be provided.

Thanks for this suggestion very much. The summary figure and figure legend has been modified to make the figure more readable and meaningful.

We hope that our responses are satisfactory. Again, we express our gratitude to the reviewers and the editor for efficient handling of this manuscript. Please do not hesitate contact me, should you have any questions.

Sincerely,

Wenbo Deng, PhD

Fujian Provincial Key Laboratory of Reproductive Health Research

School of Medicine, Xiamen University

Xiang'an Road, Xiang'an District, Xiamen, 361102, China.

Phone: +86-17850568375

Email: wbdeng@xmu.edu.cn

REVIEWER COMMENTS

Reviewer #1 (Remarks to the Author):

Manuscript ID: NCOMMS-23-09237

Title: Deciphering an indispensable role of uterine epithelial SHP2 in parturition initiation at single cell resolution

Authors: Liu et al.

This is a revised manuscript by Liu et al that describes a potential role for the non-receptor protein tyrosine phosphatase Shp2 in parturition initiation. The study has been improved in particular by providing overview staining images that delineate where the luminal and glandular epithelial compartments persist in late gestation implantation sites.

Overall, however, major issues still remain. The identification of HB-EGF up-regulation between E16 and E19 does not justify the focus on Shp2. It rather appears that the Shp2 data were available prior to the single cell data, and that the post-hoc addition of the scRNA-seq data prompted the authors to make a tenuous connection to justify the use of the Shp2 model.

The study would be presented in a much stronger way if it started with the Shp2 data in Figure 3, and focused on the inclusion of scRNA-seq data from this knockout model. This part of the manuscript is also much better written. I hence urge the authors to omit the first scRNA-seq part which comes with major issues, as highlighted below.

Major comments:

1. It is still unclear what precise tissue was single cell sequenced and Figure 1a is confusing. Given what is stated in the text, the sequenced tissue should only entail the myometrial layer in Figure 1a. If that is correct, please re-draw Figure 1a to make this clear. Also, the representation of the decidua is incorrect, as this is only a thin layer overlying the placenta at this stage. The placenta and embryo should be indicated as well in the figure. These aspects could be depicted in a grey tones, and the sequenced tissue colour-coded in red (or similar). The same applies to later figures that incorporate this diagrammatic depiction.

However, more importantly, this description in the text is completely inconsistent with the data and with later drawings in Fig. 7a and the staining in Supp. Fig. 1. It would rather appear that the entire region broadly labelled as "ST", which is de facto decidua and which contains the glands and luminal

epithelium, was sequenced. This corresponds with the identified single cell clusters that show only minor contributions of muscle cells (which should be the dominant cell type if uterine tissue “after the removal of decidua, placenta and fetus” was sequenced). If that is so, the entire description of the approach in the text is wrong and/or misleading and needs to be revised.

2. Highlighted cell type-specific markers remain unverified. Notably “Epi” vs “gland” signature genes that are meant to distinguish uterine luminal and glandular epithelium, i.e. *Msx1*, *Sox9*, (*Cebpd*), *Ehf*, *Foxp1* etc, need to be validated by immunostaining to verify their cell type specificity at this stage in gestation. This is immensely important as these are described as novel signature genes capable of distinguishing these two epithelial cell types. The nomenclature of “Epi” in this context is highly unfortunate, as both cell types are epithelial in character.

3. Some of these cell-type specific markers should be applied to the *Shp2d/d* uteri to determine which epithelial layers are particularly affected.

4. The two parts of this manuscript (scRNA-seq of WT deciduae/uteri and the *Shp2* analysis) are not inter-linked, i.e. identified signature genes of specific cell types are not applied to the KOs, and vice versa the markers applied to the KOs have little or no relationship to the scRNA-seq data. This point further underpins the advice provided above that the first part of this manuscript should be omitted.

5. Please reword cell type specific expression in *Epi_0*, *Gland_10* and *prolifEpi_20*. It is unclear what these populations are meant to be, especially as they are all epithelial cell types. This point relates to comment 2 above.

6. As noted in the summary above, the shift from *Hbegf* up-regulation to investigating *Shp2* is a huge leap. The question arises, would similar phenotypes be seen on *Hbegf* deletion or ERK inhibition specifically in uterine epithelial cells. In any case, the *Shp2* phenotype is likely far more pronounced than an *Hbegf* phenotype. As such, the study would be portrayed far better if it was focused on *Shp2*, without the single cell analysis upfront that is somewhat strenuously portrayed to justify the focus on *Shp2*.

7. The *Akr1b3* data need to be shown as a separate channel, the faint red staining is invisible in the overwhelmingly green overlay.

8. Contrary to what is stated, PGF2a injection does not lead to more females delivering earlier. Please correct this statement. Furthermore, the n=5 of PGF2a injected females is insufficient. There is an inexplicably huge discrepancy in the data between Fig. 3c (*Shp2d/d* survival rate 61%) and Fig. 3n (*Shp2d/d* survival rate 40%), so a few animals more may make a huge difference.

9. Please highlight that the parturition delay is only observed in some Shp2 d/d females, even if the various genes are de-regulated in all/most of them. Thus, the epithelial Shp2 deletion is only partially critical for determining the timing of parturition.

Reviewer #2 (Remarks to the Author):

The authors have addressed all the issues raised in the initial review .

Reviewer #3 (Remarks to the Author):

The authors have adequately addressed my concerns with the earlier version of the manuscript.

Point by point response:

Manuscript ID: NCOMMS-23-09237

My co-authors join me in expressing our sincere appreciation to the reviewer's thoughtful comments. We have critically reviewed each comment and added new experiments as suggested by the reviewers. Our responses are elaborated below. The reviewer's comments are followed by our responses highlighted in blue. All changes in the revised manuscript are marked by blue.

Reviewer #1:

This is a revised manuscript by Liu et al that describes a potential role for the non-receptor protein tyrosine phosphatase Shp2 in parturition initiation. The study has been improved in particular by providing overview staining images that delineate where the luminal and glandular epithelial compartments persist in late gestation implantation sites.

Overall, however, major issues still remain. The identification of HB-EGF up-regulation between E16 and E19 does not justify the focus on Shp2. It rather appears that the Shp2 data were available prior to the single cell data, and that the post-hoc addition of the scRNA-seq data prompted the authors to make a tenuous connection to justify the use of the Shp2 model. The study would be presented in a much stronger way if it started with the Shp2 data in Figure 3, and focused on the inclusion of scRNA-seq data from this knockout model. This part of the manuscript is also much better written. I hence urge the authors to omit the first scRNA-seq part which comes with major issues, as highlighted below.

We really appreciate the reviewer's suggestion to start with the Shp2 data in Figure 3 to avoid the tenuous connection to justify the use of the Shp2 model. We agree that the presentation of whole story is much stronger than before after this modification. In the revised manuscript, we started with the phenotype of SHP2 knock out mice, then followed by the characterization of epithelium at peri-

parturition stage and the underlying mechanism. We thank for this recommendation very much.

Major comments:

1. It is still unclear what precise tissue was single cell sequenced and Figure 1a is confusing. Given what is stated in the text, the sequenced tissue should only entail the myometrial layer in Figure 1a. If that is correct, please re-draw Figure 1a to make this clear. Also, the representation of the decidua is incorrect, as this is only a thin layer overlying the placenta at this stage. The placenta and embryo should be indicated as well in the figure. These aspects could be depicted in a grey tones, and the sequenced tissue colour-coded in red (or similar). The same applies to later figures that incorporate this diagrammatic depiction.

This is a very important suggestion. As the reviewer point out that decidua is only a thin layer overlying the placenta, the placenta has been depicted in the revised diagrams.

However, more importantly, this description in the text is completely inconsistent with the data and with later drawings in Fig. 7a and the staining in Supp. Fig. 1. It would rather appear that the entire region broadly labelled as “ST”, which is de facto decidua and which contains the glands and luminal epithelium, was sequenced. This corresponds with the identified single cell clusters that show only minor contributions of muscle cells (which should be the dominant cell type if uterine tissue “after the removal of decidua, placenta and fetus” was sequenced). If that is so, the entire description of the approach in the text is wrong and/or misleading and needs to be revised.

Thanks for this suggestion. It is true that the region what we mainly focus is “Decidua” containing many distinct cell types. We feel very sorry for the misleading labeling of “ST” in Supp Fig 1 and Supp Fig 8. “ST” is replaced with “DEC” in those figures to eliminate those misunderstanding. With regard to the muscle cells, we find that the number of this population is very limited. The major reason is that the size of muscle cell after isolation is much bigger than other cells (>100 um), which

contribute to the lower ratio of muscle cells capture due to the difficulty to pass through the flow system and mixture with droplet of 10X system. This is also documented by another single cell RNA sequencing study of human myometrium (PMID:35260533, JCI Insight 2022). In mice, our recent study finds that, accompanied with the rapid growth of embryo, the myometrium become discontinued at the mesometrial site approaching parturition. There are many stromal cells distributed in-between at this place (PMID: 37720083, iScience, 2023). Additionally, stromal cell is the dominate cell type in decidua, and it is difficult to remove other cell type as much as possible ascribed to the tight connection between each other. Collectively, these reasons contribute to the higher enrichment of stromal cell and lower proportion of muscle cell.

2. Highlighted cell type-specific markers remain unverified. Notably “Epi” vs “gland” signature genes that are meant to distinguish uterine luminal and glandular epithelium, i.e. *Msx1*, *Sox9*, (*Cebpd*), *Ehf*, *Foxp1* etc, need to be validated by immunostaining to verify their cell type specificity at this stage in gestation. This is immensely important as these are described as novel signature genes capable of distinguishing these two epithelial cell types. The nomenclature of “Epi” in this context is highly unfortunate, as both cell types are epithelial in character.

We appreciate this comment very much. More uterine luminal and glandular epithelium specific genes are validated in the revised manuscript per the reviewer’s suggestion. The term “Epi” is not very precise as epithelium include both luminal epithelium and glandular epithelium. To distinguish these two distinct epithelial cells, “Epi” has been changed to luminal epithelial cells “LE”.

3. Some of these cell-type specific markers should be applied to the *Shp2d/d* uteri to determine which epithelial layers are particularly affected.

Thanks for this suggestion very much. More cell-type specific markers have been applied in *Shp2d/d* uteri in the revised manuscript. Our result showed that the luminal epithelial genes, *Napsa* and *Csf1*, were highly expressed in LE with little in

glands in both genotypes (**Figure1** as shown below). *Prss29* was specifically localized in GE in both WT and SHP2 deleted mice (**Figure1** as shown below). The expression of *Spp1* in SHP2 mice was significantly higher than WT (**Figure2** as shown below).

Figure. 1 The marker of luminal epithelium and glandular epithelium in day 19 mouse uteri.

A-B, Sm-FISH and UMAP of *Napsa* and *Csf1* in *Shp2^{fl/fl}* and *Shp2^{d/d}* mice inter-implantation sites on day 19. LE: luminal epithelium; GE: glandular epithelium.

C-D, Sm-FISH and UMAP of *Prss29* in mice inter-implantation sites on day 19. GE: glandular epithelium.

Figure. 2 Expression of *Spp1* in *Shp2^{f/f}* and *Shp2^{d/d}* mice on day 19.

4. The two parts of this manuscript (scRNA-seq of WT deciduae/uteri and the Shp2 analysis) are not inter-linked, i.e. identified signature genes of specific cell types are not applied to the KOs, and vice versa the markers applied to the KOs have little or no relationship to the scRNA-seq data. This point further underpins the advice provided above that the first part of this manuscript should be omitted. Thanks for this suggestion very much. In the revised manuscript, we start with SHP2 knockout mice to avoid the weak connection between these two parts. Please also refer to the response above.

5. Please reword cell type specific expression in Epi_0, Gland_10 and proliferEpi_20. It is unclear what these populations are meant to be, especially as they are all epithelial cell types. This point relates to comment 2 above. Thanks for this comment very much. The “Epi” has been replaced with “LE” in the whole manuscript.

6. As noted in the summary above, the shift from Hbegf up-regulation to investigating Shp2 is a huge leap. The question arises, would similar phenotypes be seen on Hbegf deletion or ERK inhibition specifically in uterine epithelial cells. In any case, the Shp2 phenotype is likely far more pronounced than an Hbegf phenotype. As such, the study would be portrayed far better if it was focused on Shp2, without the single cell analysis upfront that is somewhat strenuously portrayed to justify the focus on Shp2.

Thanks for this comment very much. We also realize that the enrichment of EGFR signaling in the epithelium is not very reasonable to transit to the following work of epithelial function of SHP2. To increase the rationale of this study, we have modified the overall flow and mainly focused on Shp2.

7. The Akr1b3 data need to be shown as a separate channel, the faint red staining is invisible in the overwhelmingly green overlay.

Thanks for this comment very much. This have been changed accordingly.

8. Contrary to what is stated, PGF2a injection does not lead to more females delivering earlier. Please correct this statement. Furthermore, the n=5 of PGF2a injected females is insufficient. There is an inexplicably huge discrepancy in the data between Fig. 3c (Shp2d/d survival rate 61%) and Fig. 3n (Shp2d/d survival rate 40%), so a few animals more may make a huge difference.

This is a very important suggestion considering the variation of mice. We also realized that the number of mice for PGF2a injection is insufficient. We increase the mice number of PGF2a injection and the sample size is updated in the revised manuscript.

9. Please highlight that the parturition delay is only observed in some Shp2 d/d females, even if the various genes are de-regulated in all/most of them. Thus, the

epithelial Shp2 deletion is only partially critical for determining the timing of parturition.

Thanks for this suggestion very much. As mentioned by the reviewer that it is inappropriate to draw the conclusion that all/most of Shp2 d/d females manifest delayed parturition. We narrow down this conclusion in our revised manuscript accordingly.

We express our gratitude to the reviewers and the editor for efficient handling of this manuscript. Please do not hesitate contact me, should you have any questions.

REVIEWERS' COMMENTS

Reviewer #1 (Remarks to the Author):

This is a much improved version of the manuscript, and I would like to thank the authors to have taken on board the previous criticisms and for having added additional confirmative experiments.